# Dynamic modeling of EEG responses to natural speech reveals earlier processing of predictable words

Jin Dou[1]*, Andrew J. Anderson[2,3,4,5], Aaron S. White[6,7], Samuel V. Norman-Haignere[8,5,1,9], Edmund C. Lalor[1,5,10]

**1** Department of Biomedical Engineering, University of Rochester, Rochester, New York, United States of America, **2** Department of Neurology, Medical College of Wisconsin, Milwaukee, Wisconsin, United States of America, **3** Department of Biomedical Engineering, Medical College of Wisconsin, Milwaukee, Wisconsin, United States of America, **4** Department of Neurosurgery, Medical College of Wisconsin, Milwaukee, Wisconsin, United States of America, **5** Department of Neuroscience and Del Monte Institute for Neuroscience, University of Rochester Medical Center, Rochester, New York, United States of America, **6** Department of Linguistics, University of Rochester, Rochester, New York, United States of America, **7** Department of Computer Science, University of Rochester, Rochester, New York, United States of America, **8** Department of Biostatistics and Computational Biology, University of Rochester Medical Center, Rochester, New York, United States of America, **9** Department of Brain and Cognitive Sciences, University of Rochester, Rochester, New York, United States of America, **10** Center for Visual Science, University of Rochester, Rochester, New York, United States of America

* jdou3@ur.rochester.edu

## Abstract

In recent years, it has become clear that EEG indexes the comprehension of natural, narrative speech. One particularly compelling demonstration of this fact can be seen by regressing EEG responses to speech against measures of how individual words in that speech linguistically relate to their preceding context. This approach produces a so-called temporal response function that displays a centro-parietal negativity reminiscent of the classic N400 component of the event-related potential. One shortcoming of previous implementations of this approach is that they have typically assumed a linear, time-invariant relationship between the linguistic speech features and the EEG responses. In other words, the analysis typically assumes that the response has the same shape and timing for every word – and only varies (linearly) in terms of its amplitude. In the present work, we relax this assumption under the hypothesis that responses to individual words may be processed more rapidly when they are predictable. Specifically, we introduce a framework wherein the standard linear temporal response function can be modulated in terms of its amplitude, latency, and temporal scale based on the predictability of the current and prior words. We use the proposed approach to model EEG recorded from a set of participants who listened to an audiobook narrated by a single talker, and a separate set of participants who attended to one of two concurrently presented audiobooks. We show that expected words are processed faster – evoking lower amplitude N400-like responses with earlier peaks

**Data availability statement:** Raw EEG data were published by Broderick et al. on Dryad https://doi.org/10.5061/dryad.070jc, doi: 10.5061/dryad.070jc. All code needed to reproduce the results are publicly available at: https://github.com/powerfulbean/DynamicTRF.

**Funding:** This work was supported by a pilot project award from the Del Monte Institute for Neuroscience (ECL, SVNH), University of Rochester. JD received support of stipend from the Del Monte Institute pilot project award. The funders had no role in study design, data collection and analysis, decision to publish, or preparation of the manuscript.

**Competing interests:** The authors have declared that no competing interests exist.

– and that this effect is driven both by the word's own predictability and the predictability of the immediately preceding word. Additional analysis suggests that this finding is not simply explained based on how quickly words can be disambiguated from their phonetic neighbors. As such, our study demonstrates that the timing and amplitude of brain responses to words in natural speech depend on their predictability. By accounting for these effects, our framework also improves the accuracy with which neural responses to natural speech can be modeled.

## Author summary

Speech is central to human life. As such, an enormous amount of research has been done in recent decades aimed at understanding how the human brain processing speech and language. Much of this work has involved quantitatively analyzing how brain responses reflect the comprehension of words. However, the bulk of this work has assumed that brain responses to words occur at a fixed interval after hearing that word. In this study, we wished to test the hypothesis that words that are more predictable – in the context of a narrative story – would be processed more rapidly. Using a novel computational analysis, we found that the timing and amplitude of the brain's response to a word is affected by the predictability of that word and the predictability of the immediately preceding word. This finding has implications for theories of language processing. Moreover, it has implications for improved analysis of brain responses to speech.

## Introduction

The human brain understands natural speech at rates of around 120–200 words per minute. A well-known signature of this process is the N400 electrophysiological brain response, which is typically revealed by recording and contrasting electroencephalogram (EEG) responses to many sentences with unexpected vs expected endings [1]. The N400 displays a characteristic spatiotemporal profile in the form of a prominent centroparietal voltage negativity (N) at around 400ms after the onset of unexpected words (relative to expected words). Over the course of thousands of studies (many of which are reviewed in [2]), the spatiotemporal properties of the N400 have been studied in great detail. In terms of spatial distribution, the N400 is known to vary somewhat – for example between N400s evoked using visual text stimuli vs auditory speech stimuli [3]. However, in terms of its temporal properties there is some debate. Specifically, while it is accepted that age [4] and disease state [5] can influence the latency of the N400, it has been proposed that, in general, the peak latency is highly consistent across studies [6]. That said, a number of studies using carefully developed stimuli suggest that the latency can vary in systematic ways. For example, in an experiment where words were presented in triplets, it was found that the N400 to the second word was delayed and amplified if it was semantically

unrelated to the first word – perhaps reflecting the fact that it was more effortful to process [7]. Somewhat relatedly, subtle shortening of the N400 peak latency has been reported as a function of word repetition, perhaps reflecting the fact that words that are repeated more often might be processed more readily [8]. Meanwhile, speech/text that is more difficult to process – either because it is acoustically degraded [9,10] or because it is delivered at higher presentation rates [11] – has been reported to produce N400s with later peaks.

So how can we reconcile these specific findings with what appears to be quite a strong consensus that the latency of the N400 is stable within subjects [6,12]? One possibility is that subtle latency differences in the responses to individual words are blurred when calculating the N400 by averaging across many sentences. It is also possible that latency shifts in the N400 only occur in the context of very particular experimental designs (e.g., word triplets, word repetitions), and do not manifest robustly in the context of more naturalistic speech stimuli such as sentences. One way to explore these possible explanations would be to use an approach that 1) does not calculate the N400 by averaging in the same way, and 2) uses naturalistic stimuli. This is the primary goal of the present study.

A secondary goal of the present study is to explore the possibility that the identification of a word might rely on the interaction between the word's predictability and the word's phonetic structure (more specifically, how quickly the word can be disambiguated from its phonetic neighbors). For example, although in general "at" is more frequent than "atmosphere", the likelihood of encountering "at" is relatively reduced in the case of "$CO_2$ in the Earth's ____"). Discovering such putative word/context-variant neural processing dynamics would be valuable for characterizing language processing in the brain, and predicting their occurrence could be key to advancing recent attempts to build accurate neuro-computational models of natural speech comprehension.

A final goal of the present study is to examine the idea that the brain's response to a word may not depend solely on the predictability of that word but may also be influenced by the predictability of previous words. While there is a long history of priming and repetition effects in the N400 [13,14], it is less clear how the *predictability* of previous words might affect the response to the predictability of the current word; this could well be due to the common practice in N400 studies of computing the cloze probability of the target words in a sentence, but not computing cloze probability for other words in those sentences, something that would be very labor intensive based on human judgements. However, as discussed further below, computing the predictability of all words in a stimulus is now tractable with the advent of deep neural network-based language models. Facilitated by these tools, we aimed to test the hypothesis that a response to the predictability of a word might be influenced by the predictability of preceding words. Specifically, we hypothesized that, if preceding words are themselves quite unpredictable, listeners might adapt to this unpredictable environment, which might diminish how their brains respond to the (un)predictability of the current word – even in the context of natural, continuous speech.

The notion of building computational models of natural speech and language processing in the brain is beginning to seem more tractable in light of recent advances in large-scale language modeling. EEG studies of natural speech comprehension have placed heavy focus on predicting the N400 using estimates of words' unexpectedness derived from language models [15–18]. In particular language models, such as GPT-2, that are explicitly trained to predict next-word identity based on prior words, have led to natural speech N400 models based on so-called lexical surprisal estimates that quantify a word's unexpectedness as the logarithmic inverse probability of that word coming next [19]. (Please note: this definition of lexical surprisal differs from the definition of lexical surprisal as the negative log of word frequency – which is an approach for quantifying word surprisal that does not account for context and would not be as appropriate for use with our narrative stimuli.) These approaches have predicted EEG data by fitting linear regression mappings (known as temporal response functions or TRFs) [20,21] based on a series of lexical surprisal estimates that are time-aligned to word onsets and displaced at intervals to account for neural response delays (e.g., a displacement of + 400ms would capture peak N400 response). The resultant profile of regression weights reliably traces the standard N400 response profile in centroparietal electrodes [18,22]. However, just like traditional contrastive EEG experiments, these linear mappings

ultimately fit a single response profile across words and have no flexibility to capture temporal differences in neural responses to different words and contexts. The current study seeks to discover evidence for word/context-specific differences in neural response times by constructing new dynamic TRF models of EEG recordings of natural speech comprehension. (Incidentally, it is worth noting here that some previous work has fit TRFs using the semantic dissimilarity of words instead of their context-based lexical surprisal [16,23,24]. Please see supplementary material for a brief discussion of this issue).

We hypothesized that the timing of the natural speech N400 electrophysiological responses will vary across stimulus words according to each word's prior lexical context. To test this hypothesis, we implemented a new "dynamic TRF" model aimed at generating word/context-specific TRFs based on natural speech EEG data and then evaluated the model's ability to predict new EEG data. The new dynamic TRF model estimates linear transformations of a canonical N400 TRF template – modulating TRF response latency, timescale and amplitude based on the lexical surprisal of each individual word in the stimulus. Lexical surprisal is computed using a modern, transformer-based large language model (GPT-2) [25]. We examined the new dynamic TRF model with EEG recorded from subjects who listened to an audiobook narrated by a single talker (11,419 words), and also subjects who attended to one of two concurrently and dichotically presented audiobooks (5,097 or 4,365 words). The results show that modulating TRF latency improves EEG prediction accuracy in the centroparietal channels that are traditionally associated with semantic processing of speech [2,16,18] and that this effect is specific to listeners paying attention to the modeled speech. In particular, we show that predictable words are processed expeditiously – eliciting early and low amplitude responses – and that the timing of a current word's response is shaped by the lexical surprisal of both the previous and current words. Additional analysis demonstrate that this effect is not explainable simply on the basis of how quickly a word could be disambiguated from its phonetic neighbors.

## Results

### EEG recordings

We used two publicly available EEG datasets to test our hypotheses [16]. The single talker dataset was obtained from 19 participants listening to about 60 minutes of a narrative audiobook (split into 20 runs of about 3 minutes each). The cocktail party dataset was obtained from 33 participants listening to 30 minutes of two narrative audiobooks presented simultaneously – one in each ear – and with 16 participants attending to one story/ear and 17 participants attending to the other story/ear. Participants were presented with both audiobooks in 30 runs each of one-minute duration and asked to answer multiple choice questions on the content of both stories [26]. Each channel of the EEG recordings was referenced to the average data from two mastoid electrodes placed behind the ears, filtered between 0.5 and 8 Hz, interpolated if it was too noisy compared with the surrounding channels (see data preprocessing section below), downsampled to 64 Hz, and z-scored.

### Analysis overview

The overarching goal of our approach is to test for differences in EEG response timing (and amplitude) to different words in different (natural speech) contexts. One straightforward way to do this would be to divide the words in our stimuli into different bins based on their lexical surprisal (as has previously been done when modeling EEG responses to the speech envelope; [27]). However, choosing the number and width of such bins is somewhat arbitrary and is sure to mask any differences that may exist within each bin [28]. Not accounting for such differences would also mean that any EEG predictions based on such a model would be suboptimal. A more ambitious approach would be to attempt to identify time-shifts and scalings for the response to every word that would better explain our EEG data. However, as mentioned above, the stimuli for each of our experiments included ~10,000 words, meaning that a standard word-by-word brute force search across different time-shifts/time-scales would be quite intractable. As such, we chose to cast the problem as one of

optimization – specifically optimizing how well we can model EEG responses to our speech stimuli – so that we could leverage modern machine learning methods to determine how the EEG response for each word varies in terms of its timing and amplitude. Specifically, we developed a novel computational method to fit dynamic temporal response functions (TRFs) to individual words within a trial. Thus, rather than modeling neural responses to all words as simply scaling linearly with their lexical surprisal while having precisely the same temporal response profile, the new dynamic TRF has the flexibility to modulate the latency and time-scaling of TRFs for specific words in specific contexts, as well as fitting an additional scaling factor for their amplitude. Importantly, our approach differs from other recently introduced dynamic time-warping for neuroscientific data [29,30]. Specifically, while those approaches aim to time warp neural data – either with respect to oscillatory cycles [29] or across trials [30] – our approach seeks to warp the specific (predicted) neural responses to the features of individual words, including those responses that may temporally overlap with each other.

Our approach begins by fitting a standard regression mapping (i.e., a standard TRF), which assumes a linear relationship between changes in particular stimulus features and the resulting EEG response. This can be written as:

$$r(t) = s(t) * h(\tau),$$

Where $s(t)$ and $r(t)$ indicate the stimulus feature and model-predicted neural response at time $t$, respectively, and $h(\tau)$ indicates the filter weight at time lag $\tau$, relative to time $t$ (please see Methods and [20] for more details). In our study – as we explain below – our primary stimulus feature of interest was the lexical surprisal of each word in context, represented as a series of impulses at word onset, with the impulse amplitude reflecting the surprisal magnitude.

While this approach has proven useful in a wealth of studies on the neural processing of speech stimuli [15,18,31,32], it explicitly assumes that the TRF, $h(\tau)$, has the same timing and same shape for every change in the stimulus feature. Of course, the approach allows for the response to change in amplitude – but it does not account for any effects that previous words might have on the response to the current word, and, again, it does not allow for any variations in the timing or shape of the response.

## Dynamic TRF overview

To allow for more flexibility in the response, our approach here is to allow the TRF model to vary for every individual word. Specifically, we allowed the TRF for each word to vary in its timing and its shape – as well as its amplitude to scale in a word-specific way. To do this, we altered the TRF model equation as follows:

$$r_i(t) = \begin{cases} a_i \cdot h\left(c_i \cdot (t - t_i - b_i)\right), & t_i < t < t_i + \tau_{\max} \\ 0 & , \qquad otherwise \end{cases}$$

$$r(t) = \sum_i r_i(t)$$

where $i$ indexes each individual word, $t_i$ indicates the onset time of the $i$th word, $\tau_{\max}$ indicates the maximum value of time lag, $a_i$ enables variations in the amplitude of the response of the $i$th word, $b_i$ enables variations in the time shift of the response, and $c_i$ allows for variation in the temporal scale (time-stretch) of the response. Importantly, the stimulus feature of interest, $s_i$ – which in the present study was the lexical surprisal of the word at time $t$– is used to compute the transformation parameter ($a_i, b_i, c_i$; as depicted in Fig 1). Here, we distinguish $s_i$ with $s(t)$ to highlight the fact that lexical surprisal was calculated for each word but not for each time point. It should be noted that we cropped both the original and time shifted TRFs to let them have the same start and end time (please see methods for details).

By allowing the TRFs to be modulated in this way, our overarching goals were: 1) To test if dynamic TRFs provide a more accurate EEG model than the standard static TRF approach. This would provide evidence for context and

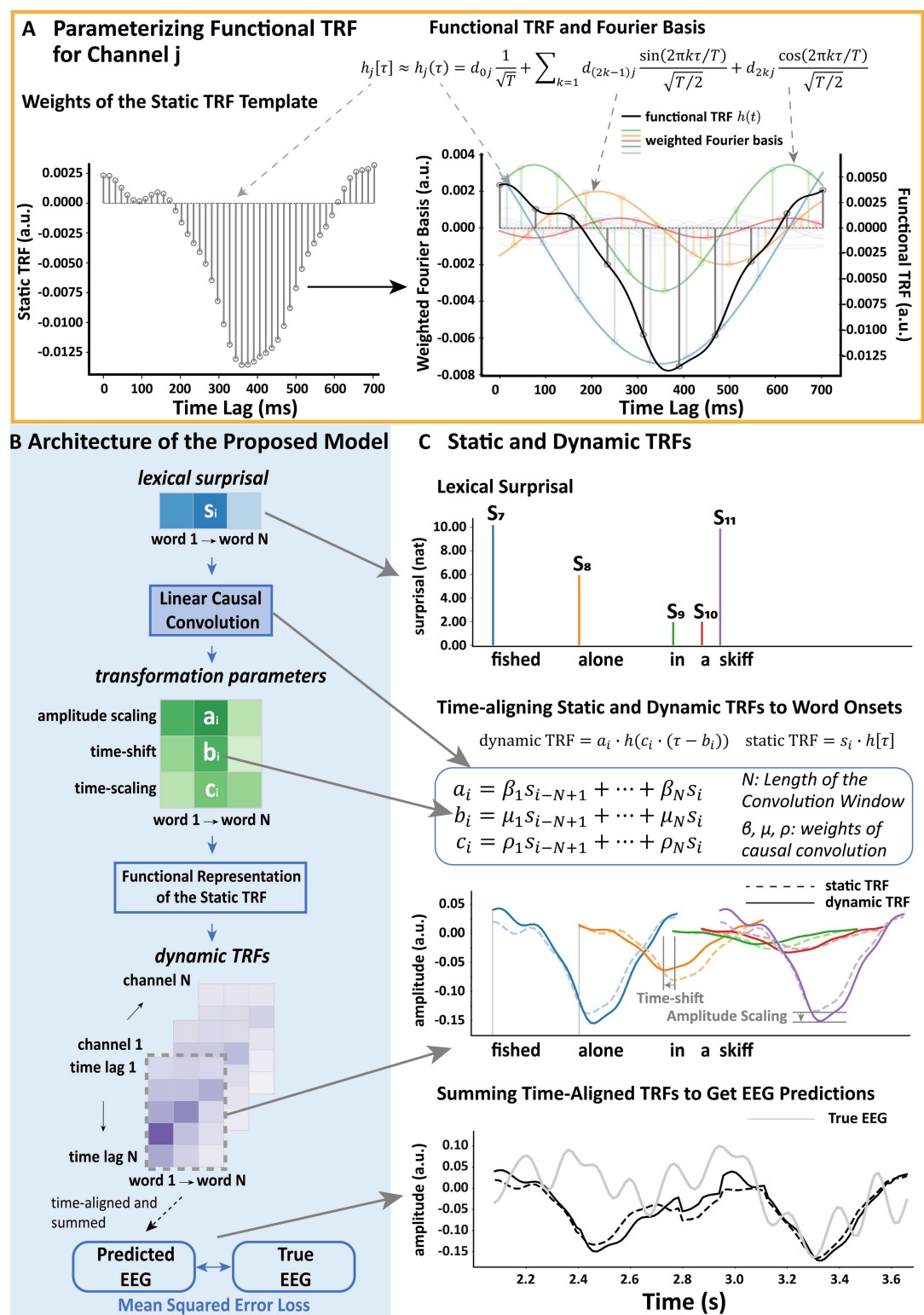

**Fig 1. Overview of the approach.** A, Representing TRF weights as a linear combination of Fourier basis functions. The left plot shows the discrete static TRF weights (canonical N400 TRF template) which were estimated using ridge regression. The right plot illustrates the functional representation of

this discrete TRF template using continuous Fourier basis functions. The functional TRF that was parameterized by the static TRF is colored in black; the weighted Fourier basis functions are in other colors. **B**, The architecture of the proposed dynamic TRF model. The lexical surprisal values of each word in our stimuli were convolved with causal kernels to estimate three transformation parameters for each word (the number of kernels defines how many transformations the model can support). The obtained transformation parameters were then used to modify the functional TRF module to get a TRF specifically tailored to each word. Specifically, the TRF for each word, $i$, could be dynamic in terms of its amplitude, $a_i$, latency (time shifting), $b_i$, or temporal scale (i.e., time stretching), $c_i$. Once these modifications had been made, the dynamic TRF for each word was aligned to that word's onset and then summed across all words to enable prediction of the EEG response. The loss we used for training the causal convolutional layer was the mean squared error between the predicted EEG and the recorded EEG. **C**, Illustration and comparison of static and dynamic TRFs estimated for words picked from the single-talker stimuli. It shows how the lexical surprisal values (top), and the TRFs (middle) were aligned to word onsets. The predicted EEG (bottom) was obtained by pointwise summing the time-aligned TRFs across words. The equation in the middle specifies how lexical surprisal was convolved with the causal kernels to get transformation parameters shown in B (green grids) for transforming the canonical N400 TRF template shown in A (left).

word-specific neural response timing. 2) To explore the relationship between EEG response timing and lexical surprisal, with the hypothesis that unsurprising words are processed more rapidly.

The dynamic TRF (Fig 1B and 1C) implements word/context-specific linear transformations of the standard static TRF. As such, the dynamic TRF begins with the computation of a static TRF – as a predictive multiple regression mapping from time-lagged stimulus features to individual electrode responses. Given that we were interested specifically in how the unexpectedness of individual words affects EEG responses, we focused our analysis on TRFs derived from the lexical surprisal of each word. Specifically, we fit static TRFs to EEG using a time series of GPT-2-based lexical surprisal impulses placed at word onsets. As we have done in previous work [32], we also included several other predictors in an effort to explain additional variance in the EEG that might derive from features that are correlated with the lexical surprisal predictor. This included one feature controlling for lexical processing, and another controlling for lower-level speech acoustics. The lexical control feature was simply a train of unit impulses marking the onset of each word. The low-level speech acoustic feature was the audio envelope of the speech stimuli. All three features were time-lagged between 0 and 700 ms. Static TRFs were fit using ridge regression in a nested cross-validation framework (see Cross Validation section). Consistent with previous research [16,22], this static TRF resembled the classic N400 event-related potential – with a marked negativity at a timelag of ~400 ms (Fig 1A).

The idea then was to modify this static TRF (in amplitude, latency (time-shift), and temporal scale (time-scaling)). To do this, we needed to learn the amplitude, time-shift and time-scaling parameters for every word. Because our stimuli have ~10,000 words, we chose to do this by exploiting machine learning methods in the context of an optimization framework. Specifically, we chose to incorporating a simple neural network (really just a single linear convolutional layer) into our framework – where we could use the error between EEG predictions and recorded EEG to learn the three parameters of interest. This involved backpropagating this error through the network to update the weights of that network (i.e., the weights that generate our three parameters of interest). To make this possible, we cannot simply use the static TRF (i.e., the average N400 shape). This is because we need to perform gradient descent in order to minimize our error and find our parameter weights. As such, we need to represent the static TRF in terms of continuous time-domain functions (Fig 1A) that can be differentiated to estimate the error gradients. Specifically, we chose to represent the TRF on each EEG channel as the linear sum of several Fourier (i.e., sinusoidal) time-domain basis functions that would capture the overall shape of the original static TRF (Fig 1A). We did this for each individual channel separately – prior to network training – via Functional Data Analysis [33]. The Fourier basis weights for each of the 128 EEG channels were then held constant for all subsequent analyses (including network training), and these basis functions could then be scaled by the three learned parameters. These three parameters were fed into the functionally represented TRF to modulate the TRF amplitude, latency (relative to word onset), and temporal scale. To reduce model complexity precisely the same three parameters were used for each of the 128 channels – thus our current model assumes that TRFs vary across different words but that this variation is shared across electrode sites.

The TRF amplitude, time-shifting, and time-scaling parameters were derived from the weighted outputs of a linear causal convolution layer – which accordingly consisted of three nodes (one for each parameter; Fig 1B and 1C). Each node performed a linear causal convolution over an ordered sequence of lexical surprisal values, with one value per word (not per timepoint). In practice, we found that convolution over only the current and preceding lexical surprisal values was sufficient to achieve maximal EEG prediction accuracy (see later analyses in S1 Fig). Consequently, the dynamic TRF's input layer contained two nodes representing the two consecutive surprisal values (unless stated otherwise) which fed into the hidden convolutional layer. Importantly, given the challenge of fitting three parameters for each word, we constrained our analysis to derive the same three parameters for all EEG channels and for all participants. In other words, we fit our parameters in a single procedure that included all channels and participants.

To predict new EEG data (Fig 1C), we constructed a time series populated by predicted TRFs by: (1) Estimating time-shifted and time/amplitude-scaled TRFs for each word. (2) Time-aligning each dynamic TRF to the corresponding word onset in a new time-series that otherwise contained zeroes. (3) Pointwise summing all new time-series across words from (2) to produce a single predicted time-series. Steps (1–3) were repeated for each of the 128 channels. To combine the new dynamic TRF-based time-series with word onset and speech envelope control predictions: (1) Time-series reflecting EEG responses to control features were predicted via a matrix multiplication of the word-onset spikes and speech envelope with the corresponding weights of the pre-computed static TRF mapping (which also supplied the N400 template). (2) The predicted control time-series were pointwise summed with the new dynamic TRF-based time-series to generate a single predicted time-series. This was repeated for each electrode. [For more details of the process used to optimize the weights of the causal convolution layer – and thus to learn the transformation parameters – please see the Methods].

## EEG responses to expected words peak earlier, but do not last longer than for surprising words

Using the single talker dataset (11,419 words), we first tested the hypothesis that modeling EEG with dynamic lexical surprisal TRFs would lead to more accurate predictions of new EEG data (without considering phonetic information for now). We reasoned that establishing improvements in EEG prediction accuracy would provide evidence that the timing of brain responses varies according to lexical context. Then, inspecting the dynamic model weights can help expose the relationship between lexical surprisal and the timing of brain responses. To recap, we had further hypothesized that unsurprising words that add little new content would be rapidly processed.

To estimate the predictive contribution of dynamically time shifting the TRF and/or scaling the TRF timescale and/or amplitude, we ran a battery of comparative analyses that modeled and predicted audiobook EEG data with the dynamic TRF, as well as reduced variants that were trained with one or two transformation parameters missing (e.g., with scale factor or amplitude scaling removed). A first observation was that we saw no predictive benefit to dynamically time-scaling the TRF (W = 25, p = 0.9986, single-tail). As such, we simplified the following presentation of results to focus only on TRF time-shift (Time), amplitude scaling (Amp) and their combination (Time&Amp).

To evaluate whether the dynamic TRF approach yielded more accurate EEG predictions than the static TRF, we deployed signed ranks tests to compare respective EEG prediction accuracies both on individual electrodes (Fig 2A), and across the entire scalp, i.e., by averaging prediction accuracies across all 128 electrodes. All EEG predictions reflected the combination of lexical surprisal with the word onset and speech envelope control features. Assessing prediction accuracy on individual channels revealed that the dynamic TRF outperformed the static TRF on a substantial number of individual electrodes (Fig 2A, Right), and that those electrodes were similar to the electrodes that show good predictive power for the static surprisal model (Fig 2A, Left). Averaging across the entire scalp (Fig 2B), we found that dynamically time shifting the TRF (Time) yielded more accurate EEG predictions than the static TRF approach (Scalp-average signed ranks: W = 147.0, p = 0.0180, single-tail). Incorporating scaling of the TRF amplitude along with the time shift (Time&Amp) yielded stronger predictions still (scalp-average W = 169, p = 0.0008, single-tail) over the static TRF.

## A  Improvement at Sites Reflecting Lexical Surprisal

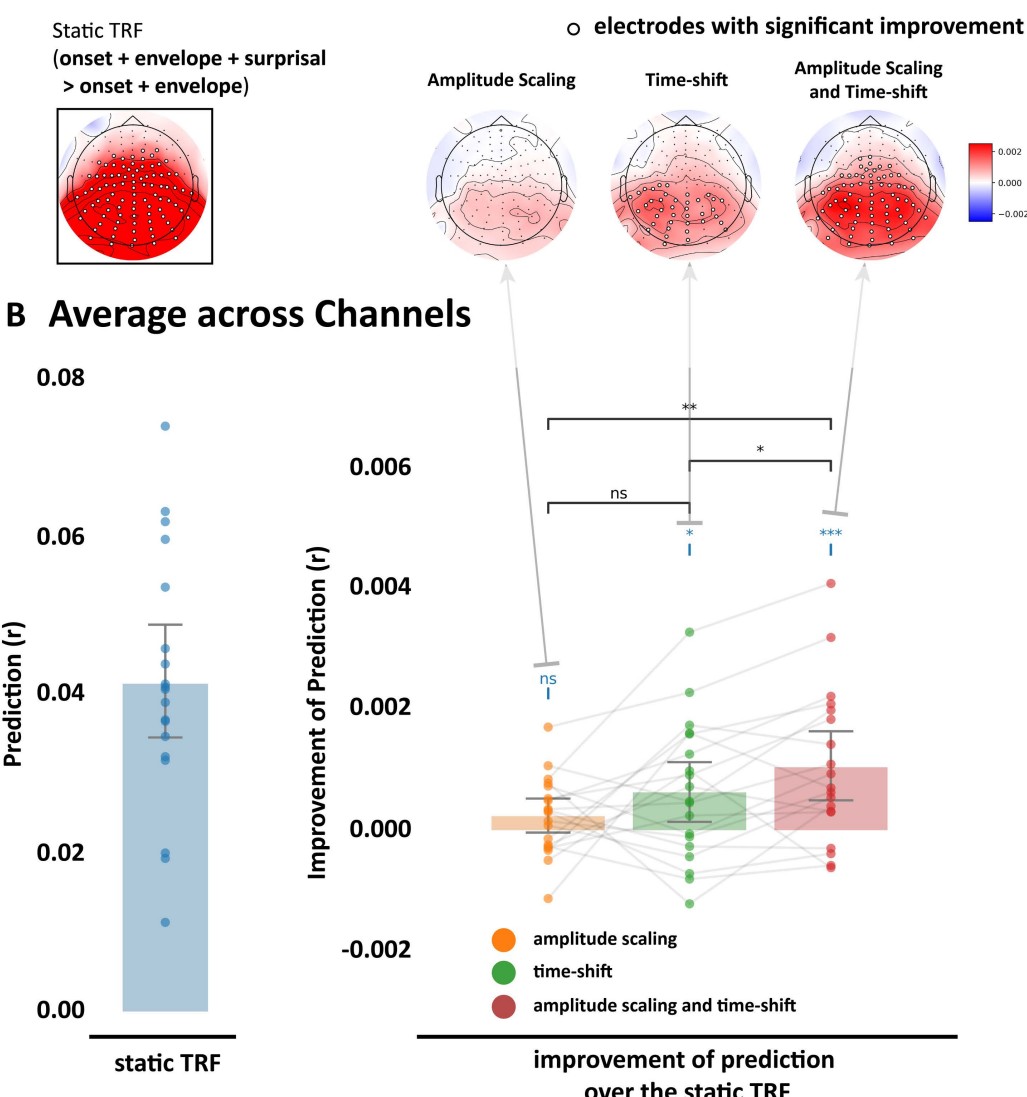

## B  Average across Channels

## C  Weights of the Causal Convolution

**Fig 2. Time-shifted TRF improves prediction accuracy.** The prediction accuracy was measured using Pearson's correlation, and the significance of improvement was measured using paired Wilcoxon signed rank tests across participants. Electrodes where the prediction improvement is significant

(p < 0.05) were labeled with white circles. P values were corrected using false discovery rate based on Benjamini-Hochberg correction (FDR), with alpha set to 0.05. **A,** Topographic maps for comparing prediction accuracies of different TRF variants on individual electrodes. The left map (in the black box) shows the improvement of prediction accuracy for a static TRF when adding lexical surprisal as a predictor in addition to the acoustic envelope and onset control predictors. The right three maps then show the additional improvement of three dynamic TRF variants over the static TRF. From left to right, the three variants are amplitude scaling (Amp), time shifting (Time), and their combination (Time&Amp). **B,** The bar plot quantifies prediction accuracies of the same static TRF (left) and dynamic TRFs (right) averaged across all 128 channels. Text/asterisks in blue in the right panel indicate statistical comparison results between each dynamic TRF and the static TRF. Horizontal connecting bars indicate the comparison between the different dynamic TRF variants. Significance is indicated by * if p < 0.05, ** if p < 0.01, and *** if p < 0.001. **C,** Convolutional kernel for the amplitude scaling (left) and time-shift (right) parameters. The lag indicates the index relative to the current time step (e.g., 0 and -1 indicate the weights for the current and previous word, respectively). Nat indicates the natural unit of information of the surprisal values. Light blue colored lines indicate the weights of models fitted from individual rounds of cross-validation (each round of which involved training an all but two runs from every participant – i.e., for each participant, two runs were left out for testing). The deep blue colored line indicates the weights averaged across those cross-validation rounds.

To then interpret the relationship between response timing and lexical surprisal (in light of the hypothesis that the brain processes expected words more rapidly), we examined the convolutional weights linking lexical surprisal input values to the time-shift node in the Time&Amp model (see Fig 2C, right; incidentally these correspond to the μ variable described in equation 10 in the methods section). Time-shift weights for the current lexical surprisal value were positive, which indeed suggests that electrophysiological responses to less predictable words (higher surprisal) peaked later than those to expected words. In contrast, time-shift weights on the previous lexical surprisal value were negative with lower magnitude. This suggests brain responses to the current word may peak earlier if the *previous* word was surprising. A similar pattern of positive/negative weightings across current and prior lexical surprisal values was observed for amplitude scaling (Fig 2C, left panel; these correspond to the β variable described in equation 9 in the methods section). Thus, if there were two surprising words in a row, the response to the latter may be earlier and shallower than if the preceding word had instead been unsurprising.

Interestingly, this interaction between the lexical surprisal of the current and previous words was crucial to the improved prediction accuracy of the dynamic TRF in the single talker condition. In particular, a dynamic TRF based on the lexical surprisal value of the current word alone produced no improvement in prediction accuracy over the static TRF (W = 124, p = 0.1289, single-tail; S1 Fig). And, incorporating the lexical surprisal of more preceding words did not lead to any systematic additional improvement.

In sum, this section provides evidence that (1) dynamic TRFs can provide more accurate EEG models than static TRFs, (2) the human brain expedites processing of expected words, and (3) that the preceding word's likelihood influences the timing and amplitude of brain responses to the current word.

## Dynamic TRFs provide more accurate models of EEG responses to attended but not unattended speech

In the previous section, we provided evidence that dynamically adjusting the amplitude and time shift of a TRF based on lexical surprisal can provide more accurate EEG predictions than the original static TRF. To confirm that this result was driven by differences in brain responses when comprehending expected/unexpected words – and not by potentially spurious correlations with the acoustics of expected/unexpected words [34–36] – we applied the same procedure as above to a cocktail party attention dataset. The cocktail party attention dataset was collected during an experiment wherein participants were asked to attend to one of two concurrently and dichotically presented audiobooks (5,097 or 4,365 words) [26,37]. We hypothesized that the dynamic TRF would produce no improvement in performance for EEG responses to unattended speech. This hypothesis was based on previous work that found little to no evidence of contextual semantic processing for unattended speech [16].

We applied the same dynamic TRF procedure – and comparison with the static TRF approach – to EEG data modeled based on both attended and unattended speech. We again chose to focus our analysis on the effect of modifying the TRF in Time, in Amp and in both Time&Amp, because we continued to observe no predictive benefit as a result of dynamically scaling the TRF timescale (attended: W = 160, p = 0.9854, single-tail; unattended: W = 266, p = 0.6043, single-tail).

We first found that the Time&Amp dynamic TRF in the attended condition yielded stronger predictions than the static TRF at a substantial number of central electrodes (Fig 3A, Left), consistent with it being an improvement in modeling responses based on lexical surprisal (since the distribution is similar to the classical N400-like topography). The dynamic TRF did not outperform the static TRF at any individual electrode for unattended speech (Fig 3A, Right). These findings were also reflected in prediction accuracies averaged over the entire scalp, with a significant improvement for the Time&Amp dynamic TRF for attended speech (W = 401, p = 0.0154, single-tail; Fig 3B), but not for unattended speech (W = 176, p = 0.9699, single-tail). In addition, directly comparing the benefit of using the Time&Amp dynamic TRF over the static TRF for attended speech with that for unattended speech revealed that the benefit was significantly greater for a substantial number of channels (Fig 3C).

Similar to single talker results, weights from the attended condition were positive and larger for the current lexical surprisal value, and smaller and more negative for the previous lexical surprisal value – both for amplitude scaling and time shifting (Fig 3D). This again indicates that more surprising current words lead to later and larger responses, with more surprising previous words reducing those effects. As was the case for the single talker dataset, the interaction between the lexical surprisal of the current and previous words was essential for the improved predictive power of the dynamic TRF (S1 Fig). Fitting such a dynamic TRF based on the lexical surprisal value of the current word alone produced no improvement in prediction accuracy over the static TRF for the attended speech (W = 242, p = 0.7545, single-tail) or (perhaps trivially) the unattended speech (W = 263, p = 0.6245, single-tail). One caveat here: we saw improved predictions for the single talker condition when using dynamic TRFs based on the current and previous word (i.e., with a window length of 2), but also for window lengths of 3, 4, 7 and 9. For the attended speech condition, we only saw the effect for a window length of 2. As such, the result must be interpreted with some caution. The fact that the pattern of weights was so similar for the attend speaker condition and the single talker condition gives us some confidence in the finding. It is also worth noting that we found no significant effect for any window length for the unattended speech. It may be that the signal-to-noise ratio of the TRFs in the cocktail party data was lower than that for the single speaker data, meaning we only saw the effect for a window length of 2.

To sum up, this section corroborates the results of the last section, by showing that the dynamic TRF provides a more accurate model of EEG responses to natural (attended) speech. Secondly, it reveals that this improvement is selective to attended speech, and is not detectable when unattended speech is modeled and the brain disengages lexical processing.

### Time shifting and amplitude scaling covary with lexical surprisal

In the last two sections, we observed similar profiles in kernel weights assigned to time shifting and amplitude scaling (Figs 2C and 3B) for (attended) speech. In both cases, the current surprisal value has a strong positive weight and the previous surprisal value has a weaker negative weight. This suggested that the variation in both time shifting and amplitude might be similar, and therefore that our network might be simplified by reducing the convolutional layer to a single variable modulating both time shifting and amplitude scaling. Because time shifting and amplitude modulation operate on different scales, the model consisted of a single convolutional operation, whose amplitude was then separately scaled for each output using a learned scalar variable (optimized in the same way as the convolutional kernel).

We hypothesized that the simplified model would be no less accurate in predicting single talker and cocktail party EEG data. This would support the idea that the time shifting and amplitude scaling effects are correlated and, thus, carry similar information. Indeed, we found that there was no significant difference in EEG prediction accuracy between the original Time&Amp model and the shared-weights variant for either the single talker (N = 19, W = 94, p = 0.5235, single-tail) or the attended speech (N = 33, W = 317, p = 0.2625, single-tail) data.

To confirm the shared convolutional layer also has a similar pattern of weights to the results shown in previous sections, we examined the weights of the shared causal convolutional layer (Fig 4A). Similar to the pattern in the previous sections, the weights estimated for both the single talker and attended speech conditions had a negative but smaller

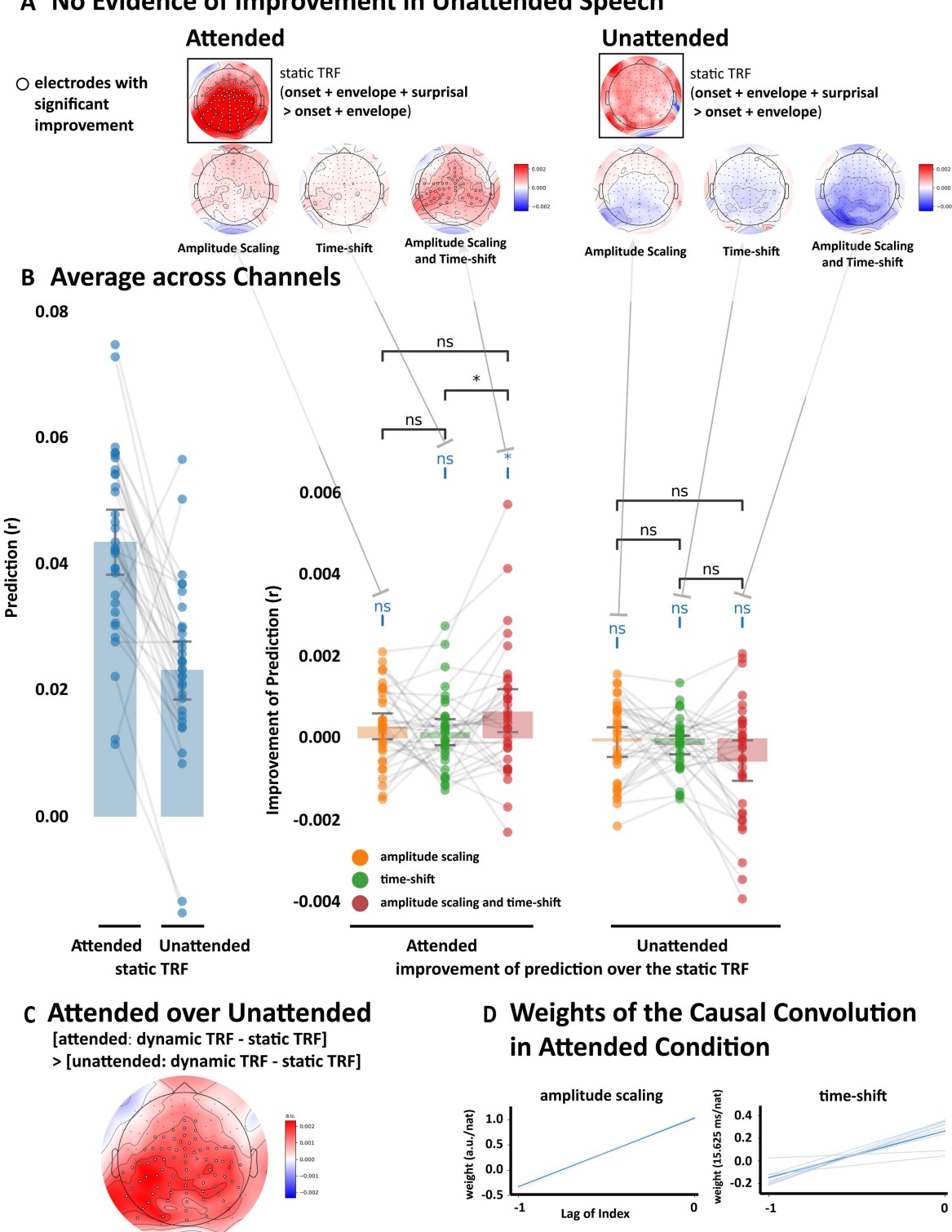

**Fig 3. The dynamic TRF improves prediction accuracy significantly only for attended speech.** The prediction accuracy was measured using Pearson's correlation, and the significance of improvement is measured using paired Wilcoxon signed rank tests. Electrodes where the prediction improvement is significant ($p < 0.05$) were labeled with white circles. P values were corrected using false discovery rate based on Benjamini/Hochberg

correction (FDR), with alpha set to 0.05. **A,** Topographic maps for comparing EEG prediction accuracies using different variants of the dynamic TRF. Same as Fig 2A, maps in the black box quantify the electrodes where including lexical surprisal as a predictor leads to significant improvements in EEG prediction under the static TRF in attended and unattended speech, respectively. Other maps on the top quantify the sites where different variants of dynamic TRF have improvement over the static TRF in both conditions. **B,** Bar plots for comparing EEG prediction accuracies using different variants of the dynamic TRF. Same as Fig 2B, the blue bar plots (left) show the prediction accuracy of the fixed TRF for attended and unattended speech. The other bar plots (right) show the improvement of prediction accuracy across the entire scalp (i.e., averaged across all channels). **C,** Directly comparing the benefit of using the Time&Amp dynamic TRF over the static TRF for attended speech with that for unattended. **D,** The weights of the convolutional layer for the amplitude scaling and time shifting. The lag of index indicates the index relative to the current time step of convolutional weight (e.g., -1 indicates the weight for the last word and 0 indicates the weight for the current word). Nat indicates the natural unit of information. Light blue colored lines indicate weights of models from individual folds of cross-validation. The deep blue colored line indicates the averaged weights across all the models.

weight for the prior lexical surprisal value, and a larger and positive weight for the current lexical surprisal value. We also examined the relationship between surprisal and time shift for each word directly (Fig 4B). The time-shift values were obtained by applying the fitted Time&Amp models on the testing data within each round of cross-validation. We also visualized how the time shift and amplitude of the dynamic TRFs varied with surprisal (Fig 4C) by randomly selecting some words along the regression line (labelled points in Fig 4B). For the single talker dataset, we visualized responses on midline parietal channel Pz, and for the cocktail party dataset, we visualized responses on the midline central channel Cz channel. Based on visual inspection of Fig 4B and 4C, it is clear that both the time shift and amplitude of the responses are larger when the lexical surprisal value of the current word are larger.

### The relationship between lexical surprisal and EEG response latency is not explained by phonetic distinguishability

As mentioned in the introduction, we were also interested in the possibility that the identification of a word might be related to how quickly that word could be distinguished from its phonetic neighbors and, indeed, how this property might influence brain responses to a word beyond any effects of the word's predictability. To explore this, we initially considered examining how a static TRF might be dynamically modified based on each word's so-called uniqueness point (i.e., the phoneme in the word where that word diverges from all other words in the language; [38]). However, on calculating this uniqueness point for each word, we discovered that it was strongly correlated (0.504 on average across all stimuli used in this dataset) with lexical surprisal (this is because longer words tend to be more informative on average; [39]). This made it difficult to disentangle the separate influences of predictability and uniqueness point on word identification. Indeed, it also caused us to worry that the lexical surprisal effect reported above might be more simply explained based on the uniqueness point of each word. To test this possibility, we built two additional models.

In the first model, we simply replaced lexical surprisal with the uniqueness point of each word as the input to each of the three nodes in the convolutional layer in the framework described above. (The uniqueness point was computed by identifying the time relative to offset of the first phoneme that enabled the word to be uniquely identified.) The idea here was that, if the uniqueness point of words explains the lexical surprisal effects we see on the dynamic TRF, then we should see very similar effects (including improved EEG prediction accuracy) when using uniqueness point as input. This turned out not to be the case. Specifically, we used signed rank tests to compare the EEG prediction accuracies between the uniqueness-point Time&Amp model and its corresponding static model, both on individual electrodes (Fig 5B Uni) and across the scalp (by averaging predictions across all channels; Fig 5A red bars). We found that the uniqueness-point Time&Amp model performed worse in both the single talker condition (scalp-average W = 9.0, p = 0.9999, single-tail) and the attended speech in the cocktail party experiment (scalp-average W = 139, p = 0.9950, single-tail). This result indicates that the uniqueness point is not the driving force behind the lexical surprisal effect reported in previous sections.

In the second model, we added the uniqueness point as a second input channel – along with lexical surprisal – to each of the three nodes in the convolutional layer in the framework described above. Thus, in this augmented model, each node in the convolutional layer received two inputs, with each input convolved with a 2-tap convolutional kernel: the

## A   Causal Convolution Weights

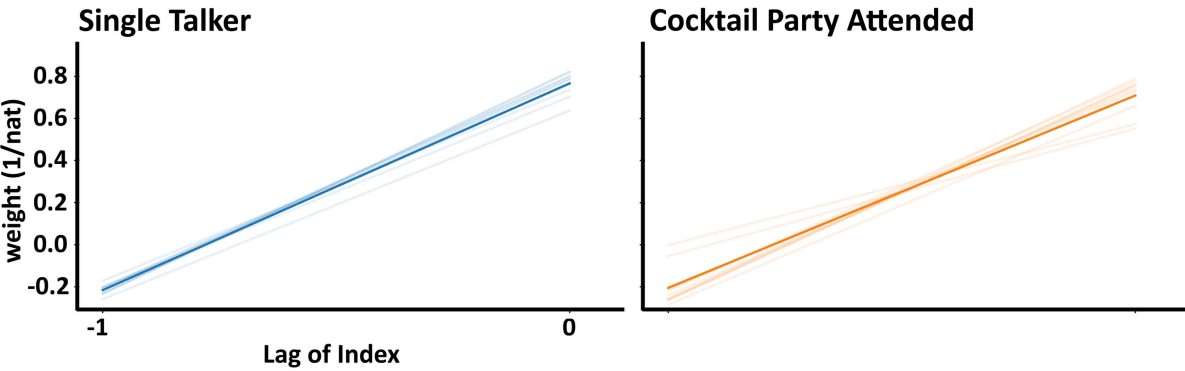

## B   Lexical Surprisal Values and Estimated Time Shift

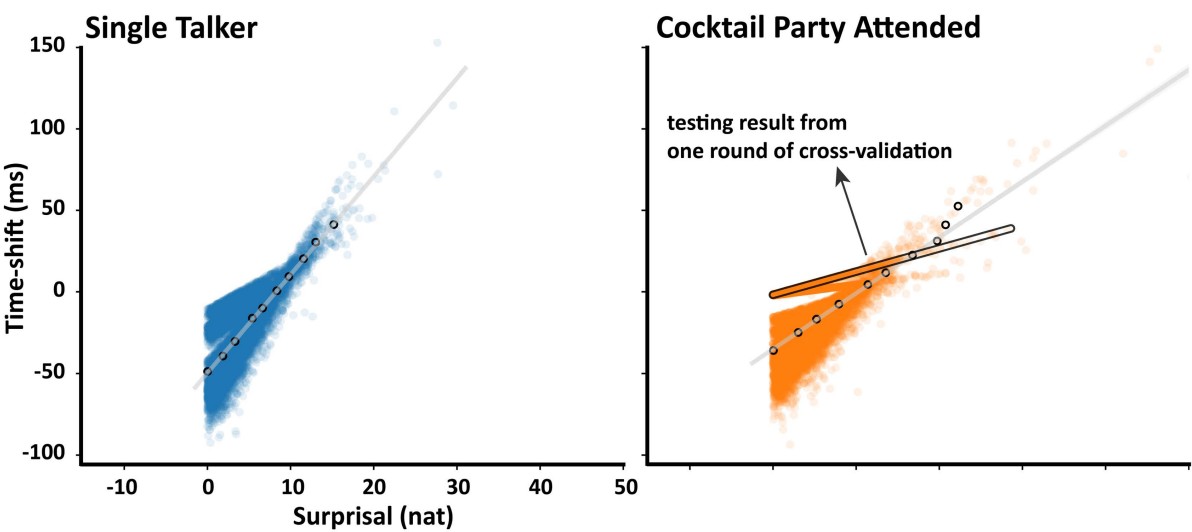

## C   Dynamic TRFs

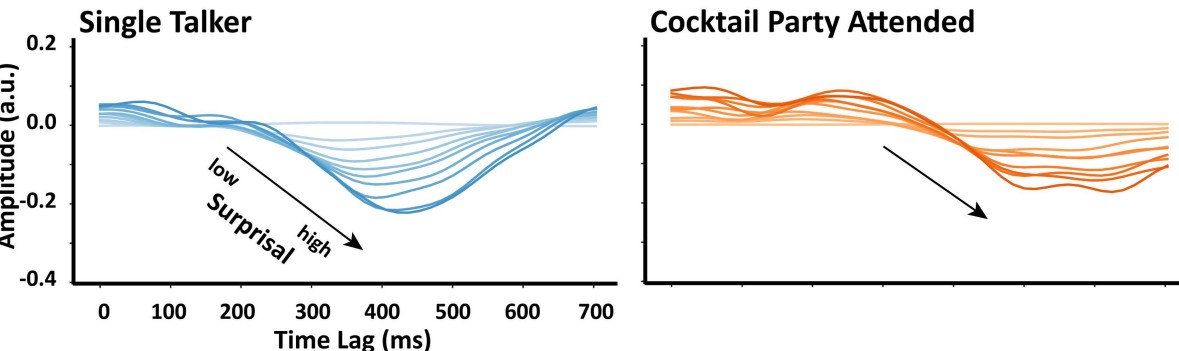

**Fig 4. Time shifting and amplitude scaling covary with lexical surprisal. A,** The weights of the convolutional layer shared by both time shifting and amplitude scaling. The lag of index indicates the index relative to the current time step of convolutional weight (e.g., -1 indicates the weight for the last word and 0 indicates the weight for the current word). Nat indicates the natural unit of information. **B,** The scatter plot of the transformation parameters. Each point indicates a pair of the estimated time-shifting parameter and the lexical surprisal value for a word from all the testing data within each round of cross-validation. A clear positive relationship is visible for the single talker and attended cocktail party speaker. **C,** Visualization of time-shifted and amplitude scaled dynamic TRFs corresponding to the selection of words indicated in **B** (the white dots). The intensity of color varies according to the value of time shift (which is equivalent to using the amplitude since they share the same output node of the convolution layer).

current and previous phonetic uniqueness point, and the current and previous lexical surprisal values. We again included word onset and acoustic envelope as co-predictors. And the model was then trained and deployed precisely as described for lexical surprisal above. The idea here was to enable us to test if the prediction accuracy of this augmented model (which included both lexical surprisal and the uniqueness point of words) was better than the prediction accuracy of the first model (which included only the uniqueness point of words). Such improved performance would be further evidence that lexical surprisal is driving the effects on the dynamic TRF, independent of the uniqueness point of words. This turned out to be the case. Specifically, we used signed rank tests to compare the EEG prediction accuracies between the augmented model and the uniqueness-point only model, both on individual electrodes (Fig 5B Sur+Uni>Uni) and across the scalp (by averaging predictions across all channels; Fig 5A right). We found that lexical surprisal improves the prediction performance significantly in both the single talker condition (scalp-average W=181.0, p=0.0001, single-tail), and the attended speech (scalp-average W=482.0, p=0.0001, single-tail). This result shows that the lexical surprisal effects are not entirely explainable based on phonetic uniqueness point.

Incidentally, to represent the uniqueness point of each word, we initially considered placing unit height impulses at the offset of the first phoneme that enabled the word to be uniquely identified. However, it was practically far easier – and just as valid – to place impulses at the start of each word whose amplitudes defined the uniqueness point of the word, i.e., words with later uniqueness points have higher amplitude impulses at word onset. The reason this was practically easier is that it more naturally fit into the framework we had developed for lexical surprisal. The reason it is just as valid is that – if there is indeed a difference in TRF shape for words with earlier vs later uniqueness points – then the framework should alter the TRF in the same way as it did for surprisal.

Finally, for completeness, we also sought to evaluate the effect of adding the phonetic uniqueness point information beyond the original lexical surprisal only model. Specifically, we used signed rank tests to compare the EEG prediction accuracies between the surprisal Time&Amp model and the augmented model that included uniqueness point, both on individual electrodes (Fig 5B Sur+Uni>Sur) and across the scalp (by averaging predictions across all channels; Fig 5A left). We found that the phonetic-augmented Time&Amp model performed better in the single talker condition (scalp-average W=160.0, p=0.0036, single-tail). This supports the possibility that lexical surprisal and the uniqueness point of a word both influence the latency and amplitude of the brain response. However, when looking across the entire scalp, this result did not replicate for the attended speech in the cocktail party experiment (scalp-average W=300.0, p=0.3688, single-tail), perhaps as a result of a slightly lower SNR in the EEG data from a multispeaker experiment. Based on visual inspection of Fig 5B, it is clear that, in the single talker condition, the uniqueness point-augmented model yielded better EEG predictions on centro-parietal electrodes (Fig 5B top right), which are the same electrodes that reflect lexical surprisal using the standard static TRF (Fig 2A left). Although there were no such predictive benefits in the attended condition across the entire scalp, some individual electrodes over centroparietal scalp showed a significant improvement after adding the phonetic uniqueness point information (Fig 5B bottom middle).

In sum, lexical surprisal adds significant value to the dynamic TRFs even when accounting for the uniqueness point of words. Considering the uniqueness point of words alone does not add predictive value to the dynamic TRF beyond the static (surprisal) TRF – although the partial improvements seen when adding both lexical surprisal and uniqueness point of words as inputs appears to provide some weak support for the idea that both features contribute to influence the EEG response to words. However, we do not wish to be too strong on the contribution from uniqueness point – given that it failed to add predictive value on its own and that it is significantly correlated with the lexical surprisal, which we know influences the responses.

## Discussion

Our study suggests that when listening to natural speech, the response to the lexical surprisal of a word, based on its preceding context, is not rigidly time-locked to word onset. By allowing a traditional static temporal response function to

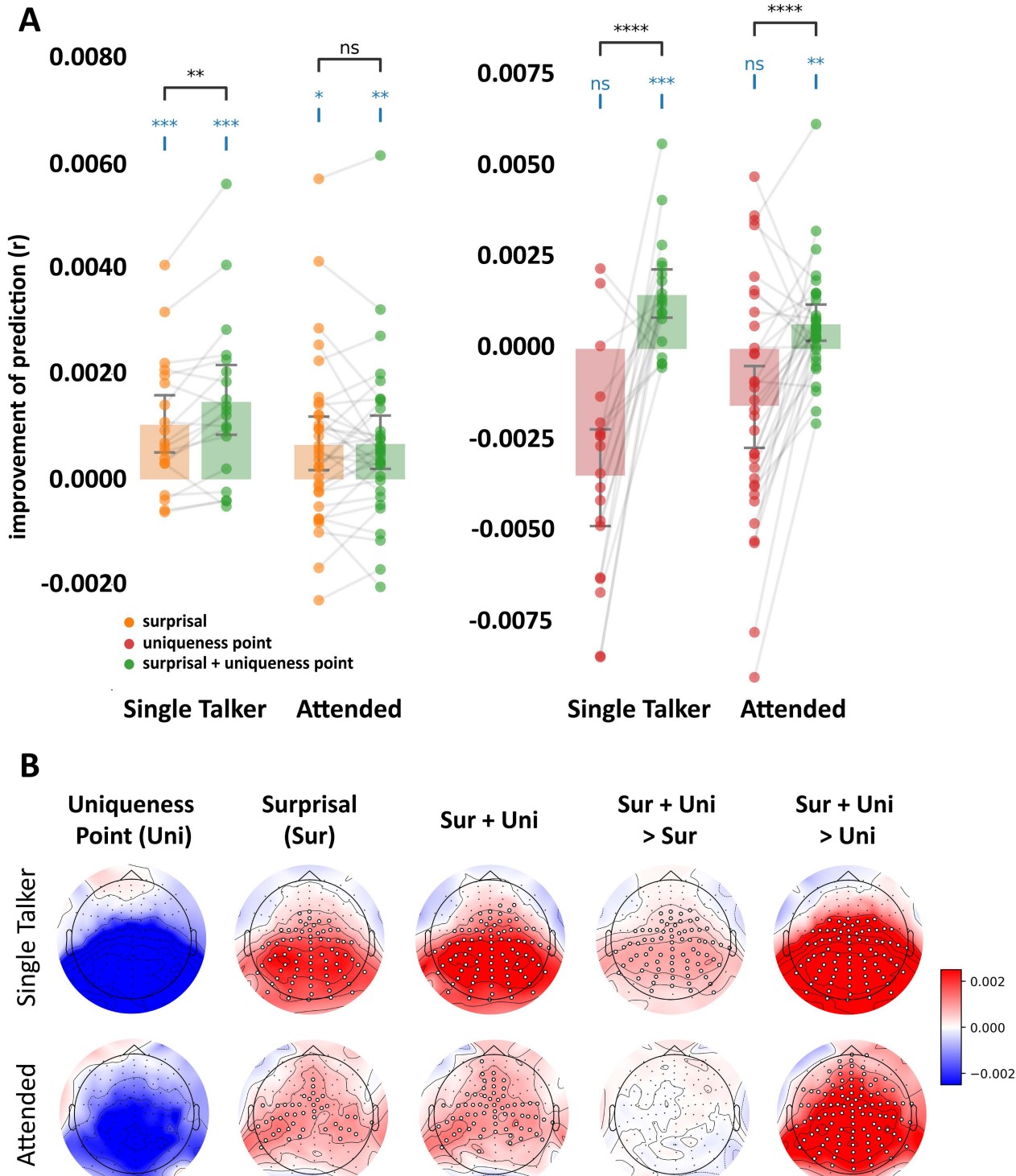

**Fig 5. Lexical surprisal has a similar impact on N400 TRF latency and amplitude when controlling for the phonetic uniqueness point of words.** **A**, Bar plots comparing the improvement in EEG prediction accuracy – relative to the static TRF – of the Time&Amp dynamic TRF models that are modulated by surprisal (orange), uniqueness point (red), or a linear combination of both (green). The uniqueness point indexes the timepoint at which a word

becomes phonetically disambiguated from other competing words. Note that the green bars in the left and right bar plots are referenced to the same data. **B**, Topographic maps for the improvement of the Time&Amp dynamic TRF with surprisal (Sur), uniqueness point (Uni), or the linear combination of both (Sur + Uni) over the static TRF (the left three columns); and the differences between the Sur&Uni and the Sur dynamic TRF, or between the Sur&Uni and the Uni dynamic TRF (the right two columns).

vary nonlinearly in terms of its time shift and its amplitude (Time&Amp), we found that both the amplitude and latency of the response to a word are influenced by the predictability of that word and the immediately preceding word. We demonstrated these finding by showing that a Time&Amp dynamic TRF model could better predict EEG collected under different conditions, relative to a standard, static TRF. The results show that, (1) dynamic TRFs can provide more accurate EEG models for lexical surprisal values than static TRFs, (2) the human brain expedites processing of expected words by reducing both the response latency and response amplitude, (3) the preceding word's likelihood influences the timing and amplitude of brain responses to the current word in an opposite direction. We also confirmed that this result was not explainable simply as a result of how quickly a word could be distinguished from its phonetic neighbors.

Our primary finding that the peak latency of the dynamic TRF correlates with the lexical surprisal of words fits with our original hypothesis that individual words may be processed more rapidly when they are predictable. This hypothesis was motivated by years of behavioral research on the role of prediction in language [40], which suggests that unsurprising words are more easily identified [41] and read more quickly [42]. However, our finding also conflicts with the near consensus in the literature that the latency of the N400 is "strikingly stable" [6,12]. That said, there is some support for N400 latency shifts in previous studies. This includes delayed N400s to the second word in a triplet if that word was semantically unrelated to the first word [7]. Subtle shortening of the N400 peak latency has also been reported as a function of word repetition [8]. Indeed, the word subtle here can also be used to describe our findings. In particular, while we did find a significant influence of lexical surprisal on peak latency (Fig 4B), the quantitative impact that this shift had on our ability to model EEG responses to speech – while also significant – was very modest in terms of effect size (Figs 1A, 1B, 2A and 2B). This suggests that latency shift may occur as a function of word predictability, but those shifts are small and may be difficult to identify in the context of N400s that are averaged over a limited number of trials. Here, by using stimuli with many thousands of words, we may have been able to pick up on this subtle effect.

Our analysis of the pattern in the convolutional weights for single talker and attended cocktail party speech revealed that those weights were positive for the lexical surprisal of the current word and negative with lower magnitude for the lexical surprisal of the previous word. This indicates that, when the current word is difficult to predict, it produces a larger, later response. However, this effect is reduced when the previous word was also difficult to predict. Indeed, a dynamic TRF based on the lexical surprisal value of the current word alone produced no improvement in prediction accuracy over the static TRF (S1 Fig). This suggests that the influence of current word's surprisal on the EEG depends on the surprisal of the previous word – with the time delay and amplitude increase being most pronounced when there is a relatively large surprisal for the current word after hearing a previous word that was not surprising. In order to learn this relationship, our model needed access to the surprisal values for both the current and previous word. This might explain why a relationship between word predictability and response latency has not previously been reported for the classic N400 ERP component – because the focus of those studies is typically on the predictability (cloze probability) of a single word. That said, the relationship we found between surprisal values and EEG response amplitude is consistent with the fact that the amplitude of the classic N400 tends to diminish with repeated presentations of surprising words [43].

Another initial goal of our study was to explore the possibility that, due to the phonetic structure of language itself, we might expect different words to be recognized at different latencies (e.g., consider "at" vs "atmosphere"). For example, it has been shown that listeners are faster in deciding whether a spoken item is a word (vs. a nonword syllable) when its uniqueness point is early in the word [38]. This was not straightforward in our dataset because of a strong correlation between the uniqueness point and the lexical surprisal of each word. However, a series of analyses

suggested that our primary finding of an influence of lexical surprisal on the timing and amplitude of EEG responses to words could not be explained as a result of correlations with the uniqueness points of words. Specifically, providing the uniqueness point of words as a lone input to the dynamic TRF pipeline did not improve EEG response predictions beyond those based on a static TRF. Moreover, dynamically adjusting TRFs based on using the combination of both lexical surprisal and the uniqueness point of words significantly outperformed a model based only on the uniqueness point of words. However, we also found that a combined model outperformed a model based only on the lexical surprisal of words for one of our data sets. This suggests the possibility that the two lexical properties might both influence EEG responses to speech. That said, the fact that a dynamic TRF based on the uniqueness point of words was no better than a static TRF means that we remain circumspect on this last point. Again, because these two lexical features were correlated, interpreting these nuanced results is not straightforward. Future experiments – likely involving speech stimuli that are constructed to decorrelate the lexical surprisal of words from their uniqueness point – will be required.

As with any effort to optimize a model based on its ability to predict data, our dynamic TRF approach runs a risk of overfitting. That said, we used cross-validation as one approach to mitigate this concern. In some previous research, we have sought to further test for overfitting by validating that models fit with extra predictors, that are randomly shuffled relative to the neural data, do not also perform better [16,44,45]. However, given the computationally heavy nature of our analysis – and the number of words in our stimuli – this was not feasible. As an alternative, we validated that the improved model performance for the dynamic TRF on the single-talker dataset also held for only the attended speaker in a cocktail party dataset, and not for the unattended speaker. Indeed, qualitatively, the analysis of both the single talker and the attended cocktail party talker produced similar results in terms of: 1) improved EEG modeling accuracy; 2) effects of lexical surprisal on response amplitude and latency; and 3) the fact that the responses to words are influenced in opposite directions by the surprisal of both the current and preceding word. The fact that this pattern of results did not hold for unattended speech is additional support for the idea that this consistent pattern of findings is not simply based on overfitting. That said, absence of evidence is not evidence of absence. It may be that the lower signal-to-noise ratio of responses to unattended speech has meant that the dynamic TRF simply could not be fit well to weaker responses in that condition. Indeed, we might go so far as to say that – based on previous research – responses to the context-based lexical processing of words in an unattended speech stream tend to be close to zero [16]. Finally, it is worth noting that the specific pattern we observe on the dynamic TRF – that of a systematic shift in its peak as a function of lexical surprisal – is further evidence that our modeling results are not simply the result of overfitting. Our analysis has revealed a systematic effect of word predictability on the peak latency of the EEG response to those words. And taking that effect into account leads to improved EEG modeling relative to a static TRF that ignores the effect.

Although the proposed variable TRF model provides an easy-to-interpret tool for modeling the dynamic nature of responses to speech, there are a number of ways in which it can be improved in future work. First, given the need to derive a functional representation of the TRF (so that the neural network can perform gradient descent), the analysis is based on an approximation of the original TRF. It is possible that the dynamically adjusting more precise models of the original TRF could produce even more accurate EEG predictions. Second, in order to prevent overfitting, we chose to dynamically adjust the TRF on every channel using a single set of transformation parameters. This may be suboptimal. Future work, likely needing larger datasets, might be able to learn channel specific modulations that would provide a more accurate model of the data. Third, due to the current group-level training protocol, the modeling we have carried out here cannot capture between-subject differences. It is natural to suppose that people have idiosyncratic neural responses, especially in challenging listening conditions. Training person-specific models may be a profitable avenue for future research. Finally, the module we used for estimating the transformation parameters was itself a linear convolutional network. As such, it may not have been able to capture what might be interesting non-linear relationships between the data and the transformation parameters.

## Methods

### Data preprocessing

The datasets we used are publicly available ([46], https://doi.org/10.5061/dryad.070jc). All EEG data were acquired at a rate of 512 Hz using an ActiveTwo system (BioSemi). 128 EEG channels plus two mastoid channels were used. The EEG data were re-referenced to the average of the mastoid channels, band-pass filtered between 0.5 and 8 Hz and downsampled to 64 Hz in MNE-Python [47]. The noisy channels were detected by: (1) finding and excluding channels with EEG whose standard deviation (std) was 2.5 times larger than the averaged std of all channels, (2) finding and excluding channels (from all channels) with EEG whose std was 0.4 times smaller than the average std of the remaining channels, (3) combining channels found in (1–2) to get the noisy channels. We then interpolated those noisy channels using the spherical spline interpolation provided by MNE-Python. The obtained data was finally z-scored within each trial.

### Experimental procedure

In the single talker condition, each participant was instructed to listen an audiobook version of the "The Old Man and the Sea" read by a single male American speaker. In total, 19 participants took part in the experiment, with each listening to 20 trials lasting about 180 seconds each.

In the cocktail party condition, each participant was instructed to attend to one talker when the audiobooks "Journey to the Center of the Earth" and "20000 Miles Under the Sea" were simultaneously presented, one to each ear. The two audiobooks were read by two different male speakers. A total of 33 participants took part in the experiment, which consisted of 30 trials, each lasting about 60 seconds.

For both experiments, the segments of the audiobooks were presented in sequence, with each trial continuing from where the previous trial concluded, i.e., the storyline was preserved.

### Speech stimulus representation

As we were interested in assessing how EEG reflects the processing of the semantic processing of words, we needed to define a representation of the speech to relate to the EEG responses. Following other recent studies [18,22], we chose to do this by computing a measure of lexical surprisal for each word.

**Lexical surprisal.** Lexical surprisal quantifies how 'surprising' or how unexpected a word is. It is defined as the negative logarithm of a word's probability of occurrence (here we used the natural logarithm):

$$surprisal = -ln(probability) \qquad \text{(Equation 1)}$$

To estimate unexpectedness of a word given prior contextual information, we derived the lexical surprisal using the large language model Generative Pre-trained Transformers 2 (GPT-2).

As with other types of so-called transformer based models [48], GPT-2 is trained to predict the next word using previous context information [25]. We restricted our use of GPT-2 to such a causal implementation in the absence of any strong evidence – as yet – that humans use predictions about future words to predict the current word. In practice, GPT-2 predicts the occurrence of a token instead of a word. A token is a common piece shared by words and is the minimal unit that a transformer model uses to model natural language. The surprisal of a token is calculated using its probability as estimated by GPT-2 (which is the conditional probability based on its prior tokens):

$$Surprisal\,(token_j) = -\log\left(P\left(token_j|context_i, token_1, \ldots, token_{j-1}\right)\right) \qquad \text{(Equation 2)}$$

Here, we chose to incorporate the maximum number of tokens that GPT-2 can use, which is 1024. Thus, based on the chain rule in probability theory, lexical surprisal for each word is the sum of the surprisal estimates for all tokens making up that word:

$$LexicalSurprisal_i = \sum_{token_j \in word_i} Surprisal\left(token_j\right)$$ (Equation 3)

where $context_i$ in Equation 2 indicates the context of $word_i$, and $\{token_j \in word_i | j = 1 \ldots N\}$.

For each trial, the surprisal of the first token is set to zero. This ignores the fact that each audiobook segment continues from where the previous segmented ended. But, given the intertrial delays (with participants sometimes answering comprehension questions and/or taking breaks), we simply assumed zero context at the beginning of each trial. Punctuation marks were included in the input of GPT-2.

**Phonetic uniqueness point.** The phonetic uniqueness point estimates the time point (relative to word onset) that the word is uniquely disambiguated from the lexical cohort of phonetic neighbors that begin with same phoneme sequence. To compute this time point, we first define what is known as the phonemic cohort entropy of each word.

The Phonemic Cohort Entropy of a word at the $i$th phoneme was derived using the following equation:

$$CohortEntropy_i = -\sum_{j}^{N} p\left(word_j^i\right) * \log_2 p\left(word_j^i\right) \#$$ (Equation 4)

where N indicates the size of the cohort, and $word_j^i$ indicates the word cohort sharing the same phoneme sequence as the target word until the $i$th phoneme [49]. This measure decreases as more phonemes are heard (i.e., as fewer and fewer potential words remain available that are compatible with the phoneme sequence).

In an attempt to separate out within-word phonetic distinguishability from context-constrained word predictability, we chose here to use absolute word frequency to quantify word probability instead of the context-based predictability that derives from GPT-2. Specifically, the probability of the $j$th word in its cohort at the $i$th phoneme was derived as:

$$p\left(word_j^i\right) = \frac{freq\left(word_j^i\right)}{\sum_{j}^{N} freq\left(word_j^i\right)}$$ (Equation 5)

The word frequency information is obtained from the SUBTLEXus database [50].

Having computed phonemic cohort entropy for each word (time aligned to the end of the corresponding phonemes in that word), we identified the uniqueness point as that point where the phonemic cohort entropy stopped decreasing.

**Word onset impulse.** The timestamps of onsets for words in the audio book were obtained by applying a forced-alignment software Prosodylab-Aligner [51] on the audio and the corresponding timestamps. A unit impulse was placed in a zero vector at the location corresponding to the timestamp of word onsets in terms of a sampling rate of 64. These impulses were included in the analysis to account for variance in the EEG that might arise at word onset unrelated to lexical surprisal.

**Lexical surprisal impulses.** Sharing the same procedure as building the word onset impulses. However, instead of a unit impulse, the amplitude of the impulses here were the corresponding lexical surprisal values.

**Speech envelope.** The broadband amplitude envelope was calculated using the Hilbert transform. As with the word onset impulses, this representation was included as a predictor in the TRF to absorb variance in the EEG that might arise from acoustic variations in the speech unrelated to lexical surprisal.

### The dynamic TRF

**Overview.** The core idea behind the proposed approach is to dynamically adjust the static TRF with context-specific linear transformations (Fig 1B and 1C). Since it is difficult to manually design the formula for such a transformation, we estimated transformation parameters in a data-driven way using gradient descent [52]. Thus, to update the weights

that are used for generating transformation parameters, the process of transforming the static TRF with transformation parameters requires that that TRF be a differentiable function. Therefore, we here chose to convert the static TRF computed from the EEG data into a differentiable continuous function $h$ (Fig 1A). Once represented in this way, this functional representation of the static TRF can be transformed into the dynamic TRF as follows:

$$dynamicTRF(\tau, a, b, c) = a \cdot h\left(c \cdot (\tau - b)\right) \qquad \text{(Equation 6)}$$

where $\tau$ are time lags referenced to word onset (as in a standard static TRF), $a$ represents amplitude scaling, $b$ represents time shift, and $c$ represents time-scaling.

**The static TRF.** The dynamic-TRF begins with the computation of a static TRF before parameterizing the functional representation of this static template (Fig 1A). A static TRF is a response function which describes the human brain as a memoryless linear time-invariant system in processing natural speech. The weights of it can be solved using a standard ridge regression mapping from time-lagged stimulus features (to account for time lags within the response function) to individual electrode responses [20]. To use the static TRF to predict unseen EEG, one can simply convolve the TRF with the time-series of the speech feature of interest (e.g., lexical surprisal at word onsets). The formula of the convolution operation is:

$$r(t) = \sum_{\tau} h(\tau)\delta(t - \tau)$$

**Functional representation of temporal response function.** To perform gradient descent and fit our dynamic-TRF parameters, we needed to re-represent the static TRF in terms of differentiable functions. We built such a functional representation ($H$) for the static TRF using functional data analysis, which decomposes the target TRF template vector into a linear combination of a group of continuous time-domain basis functions (Fig 1A right). The discrete TRF template was estimated based on ridge regression using the mTRFpy library [53]. The continuous fit was performed using functional data analysis with Fourier basis (21 bases) using the scikit-fda library (a commonly used library for performing functional data analysis, i.e., the analysis of data as continuous functions rather than simply as numerical values; [54]). The Pearson's correlation between the functionally represented TRF and the original static TRF averaged across channels was $\sim 0.99$.

**The parameterization of the dynamic TRF transformation.** The parameters that were used to transform the functional representation of the TRF were estimated using an data-driven module built using PyTorch [55]. By default, the dynamic TRF has a causal convolutional layer having separate output nodes – one for each transformation parameter (Fig 6).

When assessing whether or not modulating time shift and amplitude scaling could be accomplished with just a single variable (Fig 4), we built another dynamic TRF using a slightly different architecture. Specifically, the initial convolutional layer had a single output node, which provided the input to a new linear layer that had two separate nodes one for each transformation parameter. In this way, the module learned both an amplitude scaling parameter and a time-shifting parameter (in their appropriate units), but constrained them to be dependent on surprisal in the same way.

The weights of the causal convolutional layer were optimized on the same EEG training dataset from which the static TRF template was derived. For interpretability, the network was configured to have only six trainable weights (corresponding to the three TRF transformation parameters for the current and previous words). Weights for combining the Fourier basis set to get the 128 channels of the dynamic "N400" TRF were fixed during network optimization. Network loss was computed as the Mean Square error across all electrodes, computed between the entire time-series of predicted EEG data and the corresponding real EEG data. Training was achieved using the AdamW algorithm [56] in batches of size one run (e.g., 3min of an individual participant's EEG data) repeated across 200 epochs (training stops early when the prediction accuracy stops increasing for 10 epochs). The learning rate was cycled between two boundaries (1e-3 to 1e-2) with the higher boundary reduced after each four epochs (a cycle) to reduce the difference between two boundaries by two

**Temporal response for word $i$ at channel $j$**

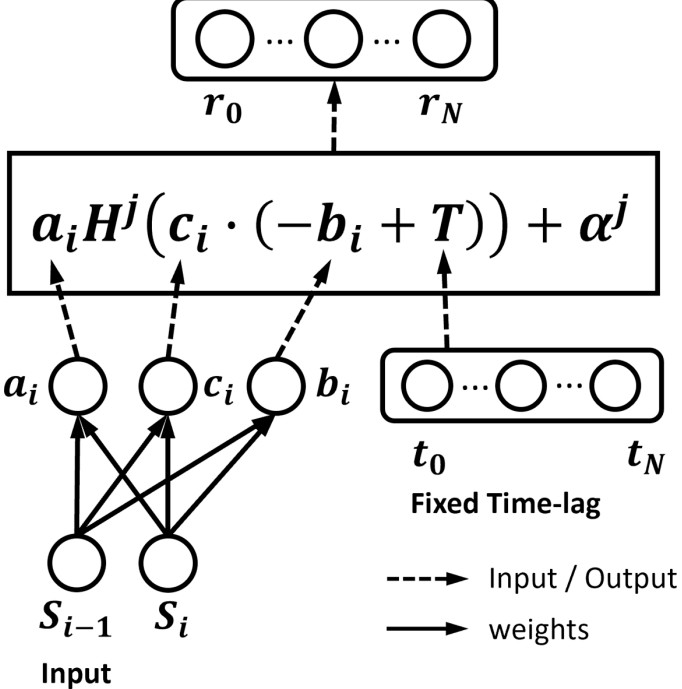

**Input**

**Fig 6. Details of the dynamic surprisal TRF model supporting time shifting, amplitude scaling, and time scaling with a window length of two.** The causal convolutional layer (left bottom) takes the lexical surprisal (or phonetic uniqueness point) of the current and previous words as input to generate the parameters for amplitude scaling ($a_i$), time shifting ($b_i$), and time scaling ($c_i$). The time-shift and time-scaling parameters were broadcast and pointwise summed with a time lag embedding ($t_0 \ldots t_N$). Given a TRF window between 0 ms to 700 ms, and a sampling rate of 64 Hz, $t_0$ will be zero and $t_N$ will be 45. The time-shifted and time-scaled time lag embedding is fed into the function TRF $H$ whose output is broadcast and pointwise multiplied with $a_i$. Then, the amplitude-scaled TRF output is broadcast and pointwise summed with the interception of channel $j$ ($\alpha^j$). The final dynamic temporal response ($r_0 \ldots r_N$) has the same length as the time lag embedding. It should be noted that we saw no predictive benefit to dynamically time-scaling the TRF, i.e., fitting the time scaling ($c_i$).

(see Training Details section) which is referred as the "triangular2" cyclical learning rate policy [57]. The weight decay (L2 regularization) was set to 1e-3. Error gradients were estimated automatically via torch.autograd.

To prevent overfitting, we used the same transformations on all time lags and channels. Thus, given the estimated transformation parameters (amplitude scaling $a_i$, time shift $b_i$, and time scaling $c_i$), the TRF for the $i$th word at the $j$th EEG channel is given by the following function (parameters colored in orange were fitted during gradient descent optimization):

$$r_{ij}(t) = \begin{cases} a_i \cdot h_j\left(c_i \cdot (t - t_i - b_i)\right) + \alpha_j, & t_i < t < t_i + \tau_{\max} \\ 0 & , & otherwise \end{cases} \qquad \text{(Equation 7)}$$

where $\alpha_j$ is the bias term of the $j$th channel of the dynamic TRF function, $t_i$ indicates the onset time of the $i$th word, and $h_j(\tau)$ is the functional representation of the TRF, which can be written as:

$$h_j(\tau) = \frac{d_{0j}}{\sqrt{T}} + \sum_{k=1}^{(N_{basis}-1)/2} d_{(2k-1)j}\frac{\sin(\frac{2\pi k}{T}\tau)}{\sqrt{T/2}} + d_{2kj}\frac{\cos(\frac{2\pi k}{T}\tau)}{\sqrt{T/2}}, (\tau = \tau_{\min}, \ldots, \tau_{\max}) \qquad \text{(Equation 8)}$$

where $d$ represents the coefficients of the basis for the TRF for the $j$th EEG channel, and $\tau_{\min}$ - $\tau_{\max}$ being the convolutional window of the TRF (reflecting the time lags between the stimulus and the brain response).Then, the transformation

parameters for the ith word – as a function of the current and $N$ previous words (i.e., a window length of $N+1$ lexical surprisal values) – can be derived using the following expressions, where, again, the terms in orange are the ones that are fit during gradient descent optimization:

$$a_i = \sum_{k=0}^{N} \beta_{N-k} s_{i-k} + \gamma_1 \qquad \text{(Equation 9)}$$

$$b_i = \sum_{k=0}^{N} \mu_{N-k} s_{i-k} + \gamma_2 \qquad \text{(Equation 10)}$$

$$c_i = \sum_{k=0}^{N} \rho_{N-k} s_{i-k} + \gamma_3 \qquad \text{(Equation 11)}$$

where $\beta$, $\mu$ and $\rho$ indicate the weights of the causal convolutional layers, with $N$ indicating the length of convolution window minus 1 and $\gamma$ indicating the bias terms. [As detailed in the main body of the paper, we found that convolution over only the current and preceding lexical surprisal values was sufficient to achieve maximal EEG prediction accuracy. In other words, we ended up using $N=1$, i.e., a window length, $N+1$, of 2.] The default value of $a_i$, $b_i$ and $c_i$ were set as $s_i$, 0, and 1 when they were not dynamically estimated by the convolution module.

In the main body of the manuscript, we noted that because the time shifting and amplitude effects appeared to covary, we might be able to simplify our dynamic TRF network by reducing the convolutional layer to a single variable, $M_i$, modulating both time shifting and amplitude scaling. Specifically, we derived the single variable $M_i$ using an equation with the same form as equations 9–11, where $\sigma$ indicate the weight of the causal convolutional layer, again the terms in orange were fit using gradient descent:

$$M_i = \sum_{k=0}^{N} \sigma_{N-k} s_{i-k} + \gamma_0 \qquad \text{(Equation 12)}$$

Because time shifting and amplitude modulation operate on different scales, we then needed to learn how to map from this single variable to the specific transformation parameters of interest. We did this by learning the orange terms in the following equations:

$$a_i = \beta M_i + \gamma_1 \qquad \text{(Equation 13)}$$

$$b_i = \mu M_i + \gamma_2. \qquad \text{(Equation 14)}$$

In terms of the time-lag parameters (i.e., the convolutional window of the TRFs), we set the $\tau_{\min}$ as 0 ms and $\tau_{\max}$ as 700 ms for the static TRF. For the dynamic TRF, we allowed a time shift of −200–200 ms. As such, the dynamic TRF could, in principle, converge on a minimum time lag of −200 ms or a maximum time lag of 900 ms. When predicting the response time series, both static and dynamic TRFs were cropped to have the same time window length of 700 ms. For a static TRF, it was always cropped to have a window of $0 - 700$ ms. For a dynamic TRF, it was cropped according to the time shift (if the time shift is −100 ms, then the cropped dynamic TRF will have a window of −100–600ms.). However, a static TRF and its corresponding dynamic TRF will be placed at the same time location in the predicted time series. In other words, the convolution window of the dynamic TRF can vary somewhat for each word, unlike the static TRF which is fixed. (Incidentally, it is important to say that, even though the convolutional window of the dynamic TRF can be longer than that of the static TRF, we do not believe this confounds our results. This is because we chose a window for the static TRF that was wide enough (0–700 ms) to more than cover any typical response to word surprisal. Allowing this to shift to include time windows from –200 ms to 0ms and from 700 ms to 900 ms should not add any extra information that could be used to

improve prediction accuracy. This is because there should be no speech related information in those additional windows. Rather, the improved prediction accuracy should come from the temporal shifting of the TRF information within the original window. Additional support for the idea that widening the window does not, in and of itself, lead to improved prediction accuracy can be seen in the lack of any improvement for the unattended speech.) Each type of transformation parameter also had a default value. For word $i$, its default amplitude scaling parameter is the same as the static TRF, which is the predictor value of that word. The default time shifting is set to 0, and the default time-scaling is set to 1. The corresponding default value is used when the type of parameter is not enabled. Instead of using the convolution formula as shown in the above section of static TRF, here we represent the dynamic TRF function as the summation of temporal responses of words:

$$r(t) = \sum_i r_i(t)$$

where $r_i(t)$ indicates the temporal response of the $i$th word.

### Accuracy metric and loss function

The loss function we used to train the model was the mean squared error between the predicted EEG and true EEG. We measured the prediction accuracy by calculating the Pearson's correlation between the predicted EEG and true EEG on each channel.

**Cross validation.** We used a standard 10-fold cross-validation procedure to test the dynamic TRF. Each dataset was equally split into 10 partitions, with each partition containing data from all the participants for 1/10 of the stimuli. Within each iteration of cross-validation, 8 partitions were used for training, 1 partition was used for validation, and 1 partition was used for testing the model. Training folds were used both to estimate the static TRF template and to train the dynamic TRF model. For the static TRF model, which is fitted using ridge regression, the validation set was used to choose the regularization hyperparameter [20,58]. For the dynamic TRF model, which is fitted using gradient decent, the validation set was used to both choose the regularization hyperparameter for the static TRF as well as decide when to stop training the dynamic parameters. Training was stopped when the prediction accuracy failed to increase for 10 epochs (an epoch indicates a single pass through the entire dataset). The partitions used for training, validation, and test were rotated across validation iterations. In detail, we first divided each story within each dataset (single talker, attended speaker, and unattended speaker) into 10 partitions. In the first round of cross-validation, we chose the EEG recordings corresponding to the first story partition as the testing set, the partition immediately subsequent as the validation set, and the remaining partitions as the training set. We did the same for the remaining 9 rounds of cross-validation, and if the last partition was used for testing, the first partition was chosen for validation.

**Statistical analysis.** Comparisons between the EEG prediction accuracies of two models to be compared (e.g., the static TRF and dynamic TRF) were calculated using the Wilcoxon signed-rank test across subjects provide by the SciPy library [59]. Since we selected the 'exact' method to calculate the signed-rank test statistics, which is recommended by SciPys when the number of samples is below 50, only W statistics were provided. Because the proposed model was hypothesized to improve the prediction accuracy, all the tests are right-tailed unless explicitly mentioned. When comparing the prediction accuracy for data on individual electrodes, the p-value obtained for each electrode were corrected using Benjamini-Hochberg correction-based false discovery rate provide by the Statsmodels library [60,61], with the alpha level set to 0.05.

### Supporting information

**S1 Fig. The effect of the window length used to predict amplitude scaling and time-shift parameters.** This figure plots the prediction accuracy (the median across subjects) for the dynamic vs. the fixed TRF models as a function of the window length used to predict the amplitude scaling and time-shift parameters of the variable TRF. Results are shown

separately for the single talker, attended and unattended conditions. The dashed line indicates the performance for the static TRF. For single talker and attended speech, the model achieved the highest prediction accuracy when its window size was around 2–3 (the improvement from 2 to 3 for single talker was not significant, $W = 133.0$, $p = 0.0668$, single-tail). For unattended speech, the prediction accuracy was always not significant compared with the accuracy of the static TRF. (TIF)

**S2 Fig. The scatter plot of the transformation parameters of the augmented model.** The scatterplot for surprisal and the time-shift. Each point indicates a pair of these two variables of a word. (TIF)

**S1 Text. Assessing the influence of convolutional window length.** (DOCX)

**S2 Text. A note on the use of context-based lexical surprisal as a predictor of EEG responses to speech.** (DOCX)

## Acknowledgments

We would like to acknowledge technical support from the Center for Integrated Research Computing at the University of Rochester.

## Author contributions

**Conceptualization:** Jin Dou, Andrew J. Anderson, Aaron S. White, Samuel V. Norman-Haignere.

**Data curation:** Jin Dou.

**Formal analysis:** Jin Dou.

**Funding acquisition:** Edmund C. Lalor.

**Investigation:** Jin Dou.

**Methodology:** Jin Dou, Andrew J. Anderson, Samuel V. Norman-Haignere.

**Project administration:** Jin Dou, Edmund C. Lalor.

**Resources:** Edmund C. Lalor.

**Software:** Jin Dou.

**Supervision:** Edmund C. Lalor.

**Validation:** Jin Dou, Andrew J. Anderson, Samuel V. Norman-Haignere, Edmund C. Lalor.

**Visualization:** Jin Dou, Andrew J. Anderson, Aaron S. White, Samuel V. Norman-Haignere, Edmund C. Lalor.

**Writing – original draft:** Jin Dou, Andrew J. Anderson, Edmund C. Lalor.

**Writing – review & editing:** Jin Dou, Andrew J. Anderson, Aaron S. White, Samuel V. Norman-Haignere, Edmund C. Lalor.

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
