## [Decision Letter · Decision Letter 0]

14 Nov 2024

PCOMPBIOL-D-24-01459Dynamic modeling of EEG responses to natural speech reveals earlier processing of predictable wordsPLOS Computational Biology Dear Dr. Dou, Thank you for submitting your manuscript to PLOS Computational Biology. After careful consideration, we feel that it has merit but does not fully meet PLOS Computational Biology's publication criteria as it currently stands. Therefore, we invite you to submit a revised version of the manuscript that addresses the points raised during the review process. Please submit your revised manuscript within 60 days Jan 14 2025 11:59PM. If you will need more time than this to complete your revisions, please reply to this message or contact the journal office at ploscompbiol@plos.org. Please include the following items when submitting your revised manuscript: * A rebuttal letter that responds to each point raised by the editor and reviewer(s). You should upload this letter as a separate file labeled 'Response to Reviewers'. This file does not need to include responses to formatting updates and technical items listed in the 'Journal Requirements' section below.* A marked-up copy of your manuscript that highlights changes made to the original version. You should upload this as a separate file labeled 'Revised Manuscript with Track Changes'.* An unmarked version of your revised paper without tracked changes. You should upload this as a separate file labeled 'Manuscript'. If you would like to make changes to your financial disclosure, competing interests statement, or data availability statement, please make these updates within the submission form at the time of resubmission. Guidelines for resubmitting your figure files are available below the reviewer comments at the end of this letter. We look forward to receiving your revised manuscript. Kind regards, Daniele MarinazzoSection EditorPLOS Computational Biology Feilim Mac GabhannEditor-in-ChiefPLOS Computational Biology Jason PapinEditor-in-ChiefPLOS Computational Biology **Additional Editor Comments (if provided):**We feel that the paper has merit and potential, but quite some work needs to be done to properly situate it in the state of the art, as well as in presenting and discussing the results.   **Journal Requirements:****Reviewers' comments:** Reviewer's Responses to Questions

**Comments to the Authors:**

Reviewer #1: Uploaded as attachment

Reviewer #2: This paper proposes that unexpected words delay EEG evoked responses and increase their amplitude. A previous study from the same lab found that the amplitude of the response at around 400 ms increased with surprisal when semantic "dissimilarity" was used as a regressor, resulting in an N400-like response. The novelty of this paper is that more surprising words also delay the response to a greater extent. However, the Discussion section does not contextualize this finding within the existing literature, making it unclear how to interpret it.

The methods used to extract changes in delay and amplitude with varying surprisals are not necessarily incorrect, but they are unclear and leave room for speculation about the actual procedure. Simpler existing methods, such as the Unfold toolbox or least squares for amplitude modulation, combined with a straightforward search for varying time delays, could have been employed. Using a general-purpose machine learning toolbox may have introduced unnecessary complexity. While this can be rectified, the authors must provide a more detailed explanation of their methodology if they choose to retain it.

The primary concern is the lack of a clear connection between the Python/machine-learning description of the process, the model equations, and the model diagram.

So in total, it is not clear that the results are new, nor is it clear that the methods are of wider interest.

More detailed comments:

Paragraph of line 219 is cryptic. I doubt it can be made much easier to understand, because the actual approach seems clunky. I doubt you will be inclined to change the approach (though that is what I would recommend). So you have no other option but to explain what you did in a way that is actually comprehensible. I tell you what I understand, and you fill in the blanks please. You fit the TRF in the time domain, rather than the time domain. Fine. Then you adjust, for each trial, amplitude, time delay and scale (shared for all EEG channels) to better fit individual trials. Why this could not have been done more simply in the time domain is not clear to me. What any of this has to do with “linear convolutional layer” and “backpropagation” I have no idea. The least-squares solution to the Fourier (or time domain) coefficients can be solved in closed form. Fitting after that the three extra parameters per trial with non-linear least squares should not be a big deal. Admittedly, adding delay and time-scales makes this non-convex nonlinear optimization, but you have a good starting point so non-convexity should not be a problem.

Maybe using ready-made python code “scikit-fda library” made the description unclear?

To gain clarity, I went to the methods (Eq. 7-15). They add little clarity. What are s_i and M_i? In Fig 1 N is the length of the convolution window and s_i are apparently the regressors. The Equations 9-15 make no sense to me.

Trying again to understand, it appears from Fig. 6 that you allow the amplitude a_i and time delay b_i to scale linearly with surprisal of the current and the previous word. Why? That really makes the interpretation of results complicated. The coefficients seem to be negative for the s_{i-1} so it looks like amplitude scales with surprisal change. Do you feel strongly about that? Then say it in the abstract.

You may also consider using the Unfold toolbox. While it is NIH (not invented here) we have used it and it really works nicely to get the different TRF (shapes) at varying amplitudes of the regressor. It has been well validated and has a clean rationale. It does give you maybe too many degrees of freedom, so you will not be able to use it single-trial, but you can at least bin into different amplitudes of the regressor making the kinds of statements you want to make. Maybe it will work for you.

The main question is if this paper is about a new method, then the methods need to be explained with much greater level of detail. If the paper is about the empirical result, it may be better to use simpler methods that make the same case without distractions.

As to results, Fig. 2, I am also confused. Of course adding single-trial parameters will improve Pearson’s r. You would have to demonstrate this on unseen data. But you can’t, as they have been fit to every trial. So some sort of shuffle argument has to be made that the observed improvement in r is larger than what is expected by chance with the extra single-trial parameters. The Wilcoxon test across subjects misses this point. Unless I am really not understanding something here. So I am not sure what to make of panels A and B. I was also confused about the lines in panel C, and only a day later did I realize that this may be the linear weights combining s_i an s_{i-1}. So we have:

a_i = beta_0 * s_i + beta_{-1} * s_{i-1} ?

If so, this can be solved in closed-form, no need for gradient descent. But maybe not, because of Fig. 4, where it seems the entire shape of the filter H (TRF) changes with surprisal.

Figure 4 B, makes sense and is interesting. A much much simpler way to show this would be to do a TRF for low and high surprisals words and show the two curves. It would tell the same message and be easier to understand. In fact, is that what Fig 4C is? If all that was done was to change amplitude and time shift, why does the entire shape of the functions change? Again, I must not be understanding something basic. By the way, that arrow is really not great. A color bar for surprisals would have been a better choice.

On the issue of “uniqueness point”, it seems clear that you want to separate that form surprisal. What is the “uniqueness” point exactly. Surprisal is coded both as a point in time and and amplitude, i.e. a pulse at word-onset corresponding to the level of surprisal. What exactly is “uniqueness point”? Also a time and an amplitude? Or just a unit-amplitude pulse that the time point where phoneme entropy no longer drops within a word?

Incidentally, should surprisal not be coded at the word offset, rather then the word-onset? How would that impact the timing results?

Minor comments:

Previously you argued that N400-like TRF is due to semantic dissimilarity. Now it is explained in terms of lexical surprisal. Same or different? Seems like worth a Discussion.

line 32: “only varies (linearly) in terms of its amplitude” linear with what? Strength of a feature? Predictability? Dissimilarity? I happen to think that most EEG amplitude correlates reasonably with prediction error (MMN, ERN, P300, N400, etc). Amplitude may not scale linearly, that is clear. The “Unfold” toolbox does a nice job with that.

Line 36: “expectedness” … hmm, you already introduced “semantic dissimilarity” last time around, now expectedness. Everyone is allowed their own words. But do we really have to introduce a new one every time? We already have “surprise”, “novelty”, “perplexity”. Those are all fine words one may want to use. You also used “surprisal” (e.g. figure 2) which seems more common in this literature. Do we really need “ expectedness” now? Since your argument seems to be about prediction, maybe say simply “prediction error”?

And actually you call it “lexical surprisals” in most places. My understanding is that this is usually defined as the neg log of word frequency in the language. Whereas you use a 1024 word context and a fancy language model, so really it is the next-word likelihood given prior context. Maybe “semantic surprisal” or “language model surprisal” would be a better term? Lexical surprisal will be confusing to most readers.

Line 564: “Time&Amp dynamic TRF”. I think you can afford the extra charters to write out these words.

Line 565 “dynamically warped TRF”, you end up just shifting and scaling amplitude. Maybe “warp” is not the right word.

**Have the authors made all data and (if applicable) computational code underlying the findings in their manuscript fully available?**

Reviewer #1: **No: ** Github is set up but empty; I assume files will be uploaded upon acceptance

Reviewer #2: None

PLOS authors have the option to publish the peer review history of their article (what does this mean? ). If published, this will include your full peer review and any attached files.

**Do you want your identity to be public for this peer review?** For information about this choice, including consent withdrawal, please see our Privacy Policy .

Reviewer #1: **Yes: ** Sander van Bree

Reviewer #2: No

 **Figure resubmission:**While revising your submission, please upload your figure files to the Preflight Analysis and Conversion Engine (PACE) digital diagnostic tool, https://pacev2.apexcovantage.com/. PACE helps ensure that figures meet PLOS requirements. To use PACE, you must first register as a user. Registration is free. Then, login and navigate to the UPLOAD tab, where you will find detailed instructions on how to use the tool. If you encounter any issues or have any questions when using PACE, please email PLOS at figures@plos.org. Please note that Supporting Information files do not need this step. If there are other versions of figure files still present in your submission file inventory at resubmission, please replace them with the PACE-processed versions. 
---

## [Decision Letter · Decision Letter 1]

24 Feb 2025

PCOMPBIOL-D-24-01459R1

Dynamic modeling of EEG responses to natural speech reveals earlier processing of predictable words

PLOS Computational Biology

Dear Dr. Dou,

Thank you for submitting your manuscript to PLOS Computational Biology. After careful consideration, we feel that it has merit but does not fully meet PLOS Computational Biology's publication criteria as it currently stands. Therefore, we invite you to submit a revised version of the manuscript that addresses the points raised during the review process.

Please submit your revised manuscript within 30 days Apr 26 2025 11:59PM. If you will need more time than this to complete your revisions, please reply to this message or contact the journal office at ploscompbiol@plos.org. Please include the following items when submitting your revised manuscript:

We look forward to receiving your revised manuscript.

Kind regards,

Daniele Marinazzo

Section Editor

PLOS Computational Biology

Daniele Marinazzo

Section Editor

PLOS Computational Biology

**Journal Requirements:**

1) Thank you for stating "All EEG data are available on Dryad at https://datadryad.org/stash/dataset/doi:10.5061/dryad.070jc"

Should your submission be accepted, we will require the following information in your Data Availability Statement:

a. The DOI provided by Dryad

b. The citation for your data package in the reference section of your manuscript

c. The citation for your data package in the methods section

2) Thank you for indicating "Regarding the copyright of Figure 1, the arrows ending with a hand come from the Adobe Illustrator," Please confirm if it was drawn using Adobe as a tool or it is a used icon directly from Adobe. 

**Reviewers' comments:**

Reviewer's Responses to Questions

Reviewer #1: Thank you for these revisions. The paper now better integrates previous work on the variability and timing effects of N400, and I thank the reviewers for responding to my concerns. My last issue is that the new introduction is written in an eccentric style that lacks clarity and structure. I think the Author Summary is quite good, and the paper would be enhanced if the introduction could adopt that same level of sharpness. I would not appeal to intuitions ("surprising", "one might expect") or rely on review/expert quotes and judgments. The writing and presentation of results could be more matter-of-fact, focusing on empirical results, theoretical arguments -- in a nutshell, more objective. The use of footnotes further suggests the writing is not as streamlined as it could be. One variation:

The overarching question is how humans process speech, N400 is a phenomenon that robustly shows up, here are experiments that show it is static (concisely describe), here are some results it is dynamic (concisely describe). Here are is why the dynamic picture may be more plausible (using frameworks, theoretical reasons, arguments, other results), here is what we do to sort out the debate, etc.

Reviewer #2: The methods are explained slightly better. But there are still things that are confusing to me:

What is the j in r_{ij}? I don’t see how that is ever summed over? But maybe one should not, because as you write in line 925, j stands for index of a “channel”. Do you mean EEG channel? But then, why is the j with the basis function in equation 8. Should it not be d_{ij}? Or do you really mean that say that every EEG channel uses its own set of basis functions?

And why not match the notation of equation 8 with Fig. 1A to help your readers understand?

Figure 6 is unchanged and remains cryptic to me. The equations talk about a scale parameter c_i but it does not show in the figure. So which is it? Do you, or do you not use a variable time scale for each word? And if you did not vary time scale, maybe that should not be in the abstract either?

By the way, for the longest time c(t-t_i) looked to me like c() is a function of time, when really it is a scalar.

Now that I understand the notion of sinusoidal basis functions, I worry about non-convex cost for the c and b parameters that are in the argument of the sin/cos functions. I guess you are fine with a local minimum? Might be worth discussing the initial conditions in these circumstances. I suppose you initialize so that c=1, b=0?

On the other hand, a_i seems linear with s_i, and the proportionality coefficients beta should be solvable in closed form. Roughly, for given h_i(t) = h(t-t_i):

beta = argmin_beta sum_t(eeg(t) – beta * sum_i s_i*h_i(t))^2 = eeg\(s*h)

But I guess, once you decide to use gradient descent, might as well do it for the quadratic term as well. Makes sense.

By the way, you still talk about “warped” in several places in your text. “warped lexical suprisal” is a particularly interesting wording. :-)

Anyway, not sure you have to do anything about any of these comments. Its your paper after all, and the message is sound, even if I found the methods hard to follow.

Hope this helped.

**Have the authors made all data and (if applicable) computational code underlying the findings in their manuscript fully available?**

Reviewer #1: None

Reviewer #2: Yes

PLOS authors have the option to publish the peer review history of their article (what does this mean? ). If published, this will include your full peer review and any attached files.

**Do you want your identity to be public for this peer review?** For information about this choice, including consent withdrawal, please see our Privacy Policy .

Reviewer #1: **Yes: ** Sander van Bree

Reviewer #2: No

**Figure resubmission:**
---

## [Editor Report · Decision Letter 2]

28 Mar 2025

Dear Mr. Dou,

We are pleased to inform you that your manuscript 'Dynamic modeling of EEG responses to natural speech reveals earlier processing of predictable words' has been provisionally accepted for publication in PLOS Computational Biology.

Best regards,

Daniele Marinazzo

Section Editor

PLOS Computational Biology

Daniele Marinazzo

Section Editor

PLOS Computational Biology

---

## [Editor Report · Acceptance letter]

PCOMPBIOL-D-24-01459R2

Dynamic modeling of EEG responses to natural speech reveals earlier processing of predictable words

Dear Dr Dou,

I am pleased to inform you that your manuscript has been formally accepted for publication in PLOS Computational Biology. Your manuscript is now with our production department and you will be notified of the publication date in due course.

With kind regards,

Anita Estes
